# The *Drosophila* EGF domain protein uninflatable sets the switch between wrapping glia growth and axon wrapping instructed by Notch

Marie Baldenius[1], Steffen Kautzmann[1], Rita Kottmeier[1,2], Jaqueline Zipfel[1,3], Christian Klämbt[1]*

[1]Institut für Neuro- und Verhaltensbiologie, Münster, Germany; [2]Dezernat Forschungsmanagement, Universität zu Köln, Köln, Germany; [3]Klinik für Psychiatrie und Psychotherapie, LVR-Universitätsklinik Essen, Essen, Germany

*For correspondence: klaembt@uni-muenster.de

## eLife Assessment

This **important** study identifies a new key factor in orchestrating the process of glial wrapping of axons in Drosophila wandering larvae. The evidence supporting the claims of the authors is **convincing** and the EM studies are of outstanding quality. After the revision, the authors have addressed most of the concerns and the manuscript has been significantly improved. Both reviewers have agreed on the significance of the work. The work will be of interest to neuroscientists working on glial cell biology.

**Abstract** In the peripheral nervous system, sensory and motor axons are generally covered by wrapping glial cell processes. This neuron-glia interaction requires an intricate coordination of glial growth and differentiation. How this is controlled molecularly remains largely unknown. At the example of *Drosophila* larval nerves, we show that glial growth, which occurs without any cell division, is initially triggered by the FGF-receptor tyrosine kinase Heartless (Htl). In a screen for genes acting downstream of activated FGF-receptor, we identified the large membrane protein Uninflatable (Uif), which supports the growth of excessive plasma membrane domains but does not support glial axon wrapping. Uif is also known to inhibit Notch. Surprisingly, we find that Notch signaling is required in postmitotic wrapping glia. While compromised *Notch* signaling results in a reduced wrapping efficiency, gain of *Notch* activity in wrapping glia leads to a hyperwrapping phenotype. Thus, Notch signaling is both necessary and sufficient for glial wrapping in *Drosophila* larvae. In addition, *Notch* suppresses both *uif* and *htl* function and thus stabilizes the switch between glial growth and glial axon wrapping. Given the general conservation of signaling mechanisms controlling glia development in mice and flies, similar mechanisms may act in the mammalian nervous system to control final glial differentiation.

## Introduction

The generation of the many cell types during development is governed by both intrinsic and extrinsic regulatory mechanisms. Extrinsic information on the status of the immediate neighbors is often conveyed by the evolutionary well-conserved transmembrane receptor Notch, which plays a decisive role in many developmental contexts (*Henrique and Schweisguth, 2019*; *Ho et al., 2020*; *Sachan et al., 2024*; *Salazar et al., 2020*). In addition, individual cell types often have to undergo discrete

developmental steps once they have been specified. In the nervous system, this can be illustrated by the example of axon wrapping glial cells. These cells must first grow to a certain size before they can begin to differentiate. How such a switch is made is not yet well understood.

Axon wrapping glial cells are generally very large cells. In the peripheral nervous system (PNS), myelin-forming Schwann cells cover large segments of a single axon. Myelin formation is directed by the axon, and only large diameter axons are wrapped with myelin. Small caliber axons, in contrast, are covered by a simple glial wrap in the so-called Remak fibers. Here, a single non-myelinating Schwann cell covers many axons. In the PNS, myelin formation is initiated by a Neuregulin-dependent activation of the EGF-receptor (*Michailov et al., 2004*; *Taveggia et al., 2005*). Interestingly, *neuregulin* mutants become normally myelinated within the CNS, indicating that oligodendrocytes have evolved an independent mechanism of myelination control (*Brinkmann et al., 2008*). An alternative pathway that might act in oligodendrocytes is initiated by the Notch receptor, which can be activated by neuronally expressed F3/Contactin, a GPI-linked membrane protein of the Ig-domain family (*Hu et al., 2003*).

Within the fly nervous system, two main glial cell types are associated with axons (*Bittern et al., 2021*; *Yildirim et al., 2022*). Within the CNS, the ensheathing glia establishes a barrier around the CNS neuropil and also wraps axons that connect the neuropil with the periphery (*Pogodalla et al., 2021*). The ensheathing glial cells are formed during embryonic development through divisions of a single glioblast that also generates all astrocytes (*Jacobs et al., 1989*; *Peco et al., 2016*). The underlying asymmetric division of the glioblast is in part controlled by *Notch* (*Peco et al., 2016*). Together with the transcription factor Pointed, Notch directs the formation of astrocytes at the expense of the ensheathing glial lineage (*Gabay et al., 1996*; *Klämbt, 1993*; *Peco et al., 2016*). In glial cells, Pointed is a target of receptor tyrosine kinase (RTK) signaling (*Klaes et al., 1994*; *Klämbt, 1993*; *Peco et al., 2016*).

Within the *Drosophila* PNS, axon wrapping is mediated by the so-called wrapping glia (*Kottmeier et al., 2020*; *Matzat et al., 2015*; *Stork et al., 2008*). It follows a similar strategy as described for vertebrate Remak fibers. Wrapping glial cells are similarly large as their vertebrate counterparts, and a single cell covers more than a millimeter of axon length (*Matzat et al., 2015*). In adult stages, excessive differentiation of glial processes can be observed around large caliber motor axons, which eventually leads to the establishment of myelin-like structures (*Rey et al., 2023*).

The development of wrapping glial cells is in part controlled by RTK signaling, which is initiated by either the EGF-receptor, the FGF-receptor, or the Discoidin domain receptor (*Corty et al., 2022*; *Franzdóttir et al., 2009*; *Kottmeier et al., 2020*; *Matzat et al., 2015*; *Shishido et al., 1997*; *Stork et al., 2014*; *Wu et al., 2017*). In the developing adult visual system, the FGF-receptor Heartless initially controls proliferation and migration of the wrapping glial progenitor cells, which upon contact to axons stop their migration to then grow in size and differentiate (*Franzdóttir et al., 2009*; *Sieglitz et al., 2013*).

Glial cell development does not depend solely on RTK signaling. In *Drosophila* embryos, proper migration of postmitotic peripheral glial cells requires *Notch* activity (*Edenfeld et al., 2007*). Additional postmitotic *Notch* functions have been reported in the adult *Drosophila* CNS. *Notch* activity is needed in olfactory neurons innervating specific glomeruli for long-term memory formation, depending on presentation of Delta by projection neurons, as well as on neuronal activity (*Ge et al., 2004*; *Kidd et al., 2015*; *Lieber et al., 2011*; *Zhang et al., 2013*; *Zhang et al., 2015*). Moreover, since *Notch* is prominently expressed by postmitotic glial cells in the adult *Drosophila* CNS, it may be involved in glial differentiation (*Allen et al., 2020*; *Davie et al., 2018*; *Li et al., 2022*; *Seugnet et al., 2011*).

We have previously shown that in peripheral larval nerves, the FGF-receptor Heartless controls glial growth (*Kottmeier et al., 2020*). To gain a deeper understanding of how differentiation of wrapping glial cells is regulated, we initiated a genetic screen looking for genes that act downstream of activated *heartless*. In *Drosophila*, such suppressor screens have been used successfully many times (*Macagno et al., 2014*; *Rebay et al., 2000*; *Therrien et al., 2000*). Our screen led to the unexpected identification of the large transmembrane protein Uninflatable (Uif), which in epithelial cells localizes to the apical plasma membrane. Loss of *uninflatable* suppresses the phenotype caused by activated RTK signaling. In addition, we found that *uif* knockdown, as well as *uif* knockout larvae, show impaired glial growth. In contrast, an excess of Uninflatable leads to the formation of ectopic membrane processes that, however, fail to interact with axons. *uninflatable* is also known to inhibit *Notch* (*Xie et al., 2012*). Indeed, we could show that canonical Notch signaling is activated in wrapping glia by

the unconventional ligand Contactin, where it is required and sufficient for axon wrapping. Moreover, *Notch* counteracts both *uninflatable* and *heartless* function. Thus, Uninflatable acts to switch the balance between glial growth induced by RTK signaling and wrapping of axons.

## Results

### The FGF-RTK Heartless triggers glial growth

The development of wrapping glial cells in *Drosophila* depends on the activity of several RTKs (*Corty et al., 2022*; *Franzdóttir et al., 2009*; *Kottmeier et al., 2020*; *Matzat et al., 2015*; *Wu et al., 2017*). Loss of the FGF-receptor Heartless specifically in wrapping glial cells causes a reduced complexity of wrapping glial cell processes in the PNS of third instar larvae (*Kottmeier et al., 2020*; *Figure 1A and B*). This is reflected in a reduced wrapping index which indicates the percentage of individually wrapped axons or axon fascicles (*Matzat et al., 2015*). While in control larvae, the wrapping index is around 0.18; it drops to 0.07 when *heartless* is specifically silenced in wrapping glia (*Kottmeier et al., 2020*).

In contrast, gain of *heartless* function that is caused by expression of $\lambda htl$ (*Michelson et al., 1998*) specifically in the wrapping glia results in exuberant glial growth but does not trigger cell division (*Figure 1C*). Segmental nerves are swollen in an area that is demarcated by the position of the wrapping glial cell nucleus. To determine axonal wrapping, we prepared third instar larvae as open book filet preparations for electron microscopic analyses. This allowed us to analyze the nerve ultrastructure at positions indicated (*Figure 1C*). Far from the nerve bulge, only little remnants of a poorly differentiated wrapping glial cell can be detected (*Figure 2A and B*). In positions closer to the bulged nerve area, the wrapping glia ramifies. However, in many cases, glial processes rather grow along each other than around axons, resulting in fascicles surrounded by extensive glial cell processes (*Figure 2C*). These glial membrane formations become more evident as the diameter of the nerve gets larger (*Figure 2D–G*). In the central area of the nerve bulge, liquid-filled areas are detected with only a few thin glial cell processes (*Figure 2H*). In conclusion, expression of activated FGF-receptor Heartless triggers glial growth but does not instruct increased wrapping of axons.

### Identification of genes acting downstream of *heartless*

In order to identify genes that are responsible for this excess in glial growth, we performed a wrapping glia-specific RNA interference (RNAi)-based suppressor screen in animals that concomitantly express activated Heartless and double-stranded RNA directed against individual genes (see *Supplementary file 1* for all genes tested). For this, we generated a stable stock that allows Gal4-based expression of activated *heartless* in wrapping glia and also allows easy scoring of the wrapping glial shape [*nrv2-Gal4/ CyO^weep^; UAS-λhtl, repo4.3-stinger::GFP/TM6*]. Virgins collected from this stock were crossed against males carrying the different *UAS-dsRNA* elements (*Supplementary file 1*). The offspring third instar larvae were assayed for the presence of nerve bulges under a UV-dissecting microscope (see Materials and methods, *Figure 1—figure supplement 1* for classification scheme). *repo4.3-stinger::GFP* directs expression of nuclear GFP in all glial nuclei. Among those, the wrapping glial nucleus can be easily identified based on the large size and typical position (*Figure 1—figure supplement 1*).

To test the efficacy of the above screening settings, we first silenced genes known to be involved in FGF-receptor signaling (*Ashton-Beaucage and Therrien, 2017*; *Brunner et al., 1994*; *Klämbt, 1993*; *Vincent et al., 1998*). The activated Heartless RTK recruits the adaptor protein Stumps, which then signals via Son of sevenless (Sos), Ras85D, members of the MAPK cascade, the MAPKKK raf, the MAPKK Downstream of raf1 (*Dsor1*), the MAPK rolled (*rl*), and the transcription factor Pointed that can be phosphorylated by Rl. Knockdown of all these genes, except *Raf*, either fully or almost fully rescued the nerve bulging phenotype (*Supplementary file 1*, *Figure 1—figure supplement 1*). Full rescue (classified as 4): *stumps*, *Ras85D*, *pnt*; almost full rescue (classified as 3): *Dsor1*, *rl*, *sos*. These findings indicate that the screening works efficiently.

We next selected 2679 genes which we tested for their ability to suppress the $\lambda htl$-induced glial phenotype (*Supplementary file 1*). The different genes were chosen based on RNAseq data, glial expression, and their involvement in major signaling pathways (*Avet-Rochex et al., 2014*; Petri and Klämbt, unpublished). The knockdown of 2106 genes did not modify the nerve bulging phenotype, while knockdown of 105 genes caused a mild rescue and the knockdown of 97 genes a moderate

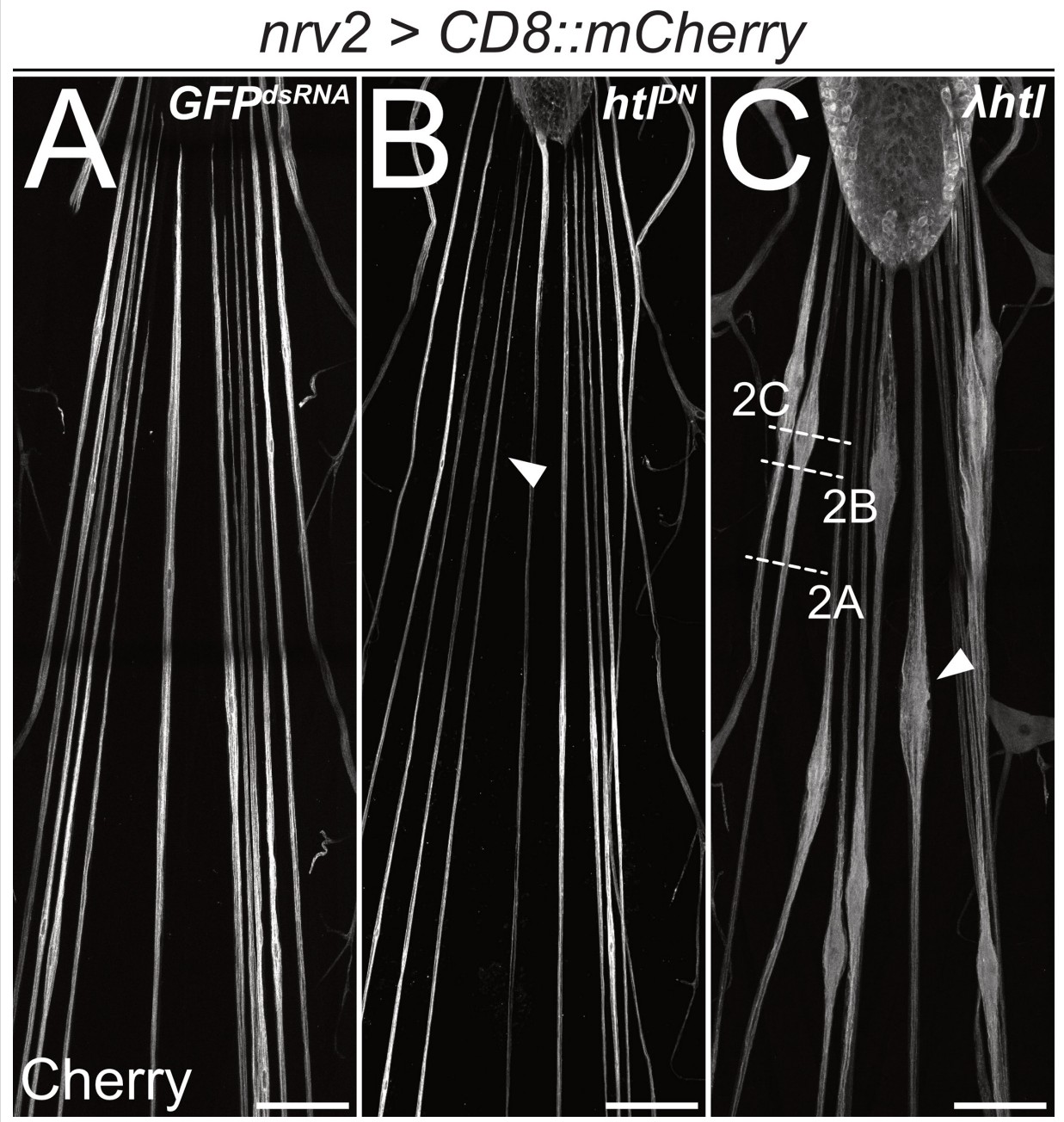

**Figure 1.** Heartless is required for wrapping glia development. Confocal images of third instar larval filet preparations, stained for CD8::mCherry expression. The segmental nerves posterior to the ventral nerve cord are shown. Wrapping glial morphology is (**A**) not changed in control animals expressing (*nrv2-Gal4*) mock *GFP^dsRNA*. (**B**) Differentiation of wrapping glial cells is affected following expression of a dominant negative form of Htl and thin wrapping glial cells are detected (arrowhead). (**C**) Expression of a constitutively active form of Htl [*nrv2-Gal4; UAS- λhtl*] leads to nerve bulges around the wrapping glial nuclei (C, arrowhead). Scale bars 100 μm. The white dashed lines indicate the level of sections shown in *Figure 2*.

The online version of this article includes the following figure supplement(s) for figure 1:

**Figure supplement 1.** Suppressor screen for genes downstream of activated *heartless*.

rescue (classified as 1 or 2, respectively, see *Figure 1—figure supplement 1*; *Supplementary file 1*). Knockdown of 318 genes caused an almost complete or full rescue of the phenotype (classified as 3 or 4, respectively, see *Figure 1—figure supplement 1*; *Supplementary file 1*). Knockdown of the remaining 33 genes caused variable phenotypes or early lethality (*Supplementary file 1*).

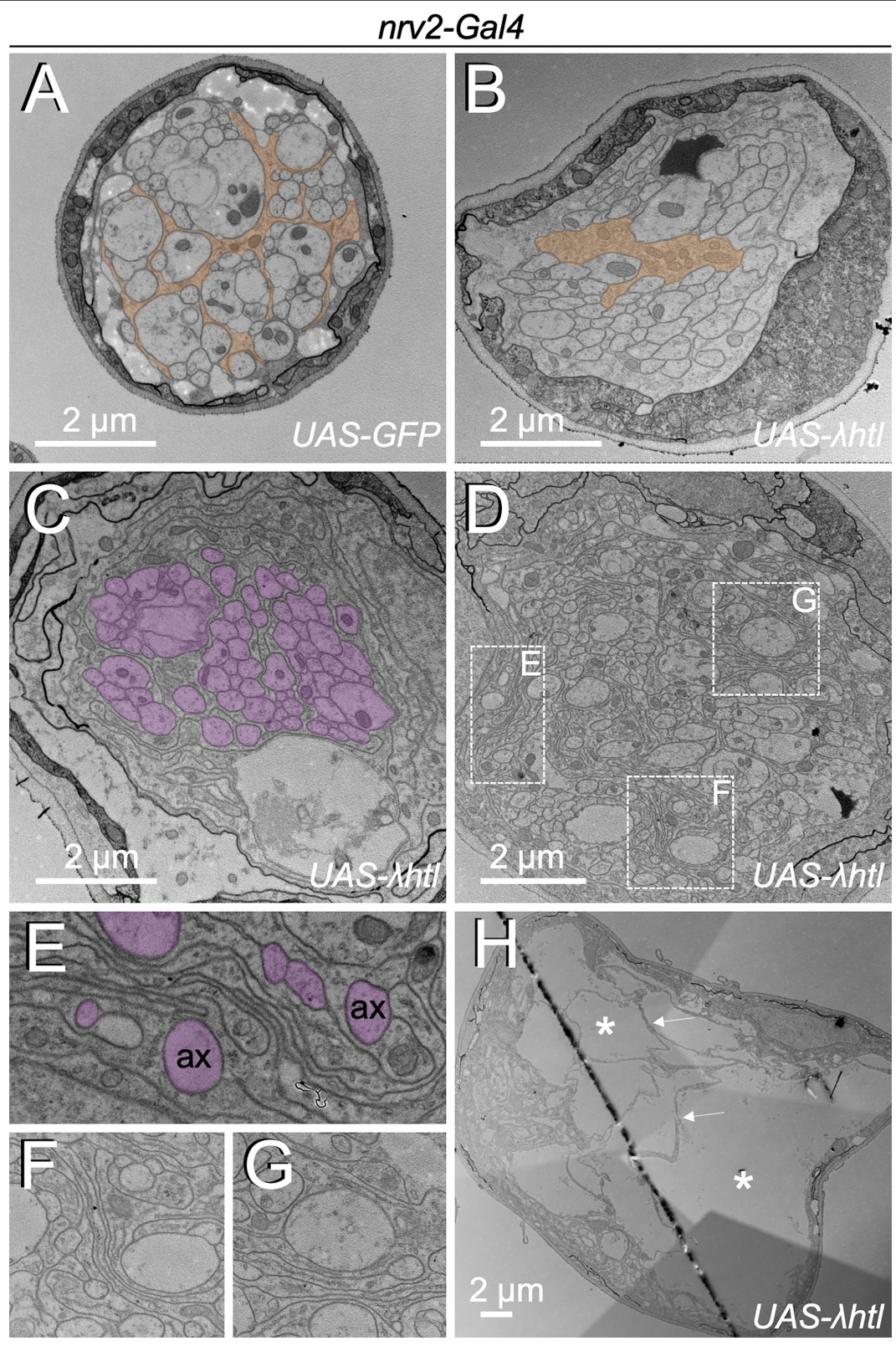

**Figure 2.** Heartless controls growth of wrapping glia cells. Electron microscopic images of segmental nerves from wandering third instar larvae. (**A**) Nerve of a control larva expressing GFP in wrapping glia sectioned 150 µm posterior to the ventral nerve cord. Normally differentiated wrapping glia can be seen. (**B–H**) Nerves of larvae expressing activated Heartless in wrapping glia [*nrv2-Gal4; UAS-λhtl*] sectioned at the positions indicated in *Figure 1C*. (**B**) Note the poorly differentiated wrapping glial cells distant from the nerve bulge. (**C,D**) At the beginning of the nerve bulge, excessive

*Figure 2 continued on next page*

*Figure 2 continued*

differentiation of wrapping glial cell processes starts to be detected that do not always grow around axons (magenta). (**E–G**) Higher magnifications of the boxed areas in (**D**). Note the formation of wrapping glial cell processes that do not contact axons (ax, magenta) but rather contact glial cell processes. (**H**) In the central area of the nerve bulge, liquid-filled vacuolar structures (asterisks) can be detected. Thin wrapping glial cell processes (arrows) span the bulged area.

We then tested additional UAS-dsRNA transgenes for the 318 genes whose knockdown efficiently rescued the nerve bulge phenotype in the initial screen. For 62 of these genes, we identified at least two independent UAS-dsRNA transgenes targeting independent regions of the mRNA that rescue the *λhtl*-induced nerve bulging phenotype (***Supplementary file 2***). For an additional 57 genes, a second but overlapping UAS-dsRNA construct was able to rescue the *λhtl*-induced nerve bulging phenotype (***Supplementary file 3***).

The above 119 candidates are likely to act in the wrapping glia. We therefore performed wrapping glia-specific knockdown experiments using the following stock [*w; nrv2-Gal4/nrv2-Gal4; nrv2-Gal4, UAS-CD8::mCherry/TM6*], where the presence of CD8::mCherry served as a proxy to determine the morphology of the wrapping glia. Indeed, knockdown of the majority of the 119 candidate genes caused wrapping glia differentiation defects (92 with phenotype, 7 no phenotype, 20 not tested, ***Supplementary files 2 and 3***). In most cases, wrapping glial differentiation was impaired, similar to what has been noted upon loss of *heartless* activity (***Kottmeier et al., 2020***). Moreover, pan-glial knockdown of most of these genes caused lethality (***Supplementary files 2 and 3***), further supporting the notion that we identified genes relevant to glial development.

Among the group of candidate genes, most encode proteins involved in translation and protein stability (48/119), transcription and splicing (23/119), lipid metabolism and membrane dynamics (20/119). This further supports the notion that a main function of FGF-receptor signaling is to promote cellular growth of the wrapping glia.

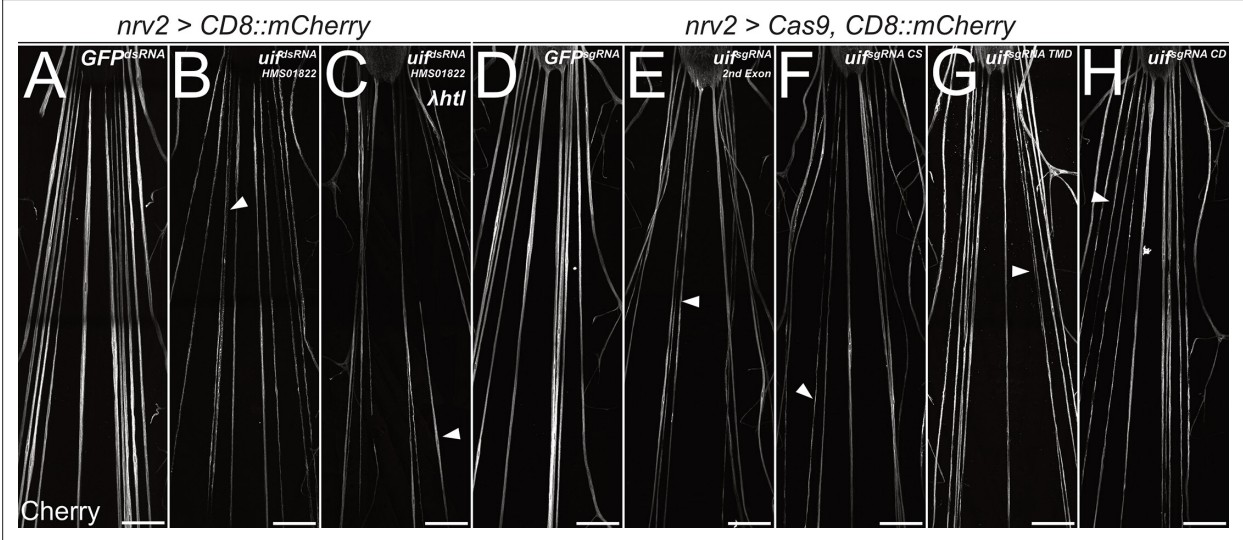

**Figure 3.** *uninflatable* affects differentiation of wrapping glia. Confocal images of third instar larval filet preparations, stained for CD8::mCherry expression. The segmental nerves posterior to the ventral nerve cord are shown. (**A**) Filet preparation of a control third instar larva. (**B**) *uif* knockdown in wrapping glial cells [*nrv2-Gal4; UAS-uif^dsRNA-HMS01822*] impairs their development, which (**C**) cannot be rescued by co-expression of activated Htl. (**D–H**) Ubiquitous expression of (**D**) mock control *GFP^sgRNA*, (**E**) *uif^sgRNA 2nd Exon*, (**F**) *uif^sgRNA CS*, (**G**) *uif^sgRNA TMD*, or (**H**) *uif^sgRNA CD* in *Cas9*-expressing larvae. All *uif* sgRNAs disrupt wrapping glial cell development, and wrapping glia appear thin and fragmented (arrowheads). n=5 larvae for all genotypes. Scale bars 100 µm.

The online version of this article includes the following source data and figure supplement(s) for figure 3:

**Source data 1.** Excel file giving the number of axons in the analyzed nerves to calculate the wrapping index.

**Source data 2.** Excel file giving the number of axons in the analyzed nerves to calculate the wrapping index.

**Figure supplement 1.** Schematic representation of sgRNA target sites used for CRISPR-mediated *uif* knockout.

## Uninflatable is required for wrapping glial cell growth

uninflatable (uif) is among the 119 candidate genes that, when silenced, suppress the nerve bulging phenotype induced by activated heartless (Figure 3A–C). It encodes a large, single-pass transmembrane adhesion protein with 18 EGF-like repeats in its extracellular domain that specifically localizes to the apical membrane domain of epithelial cells (Zhang and Ward, 2009; Figure 3—figure supplement 1). Since uif null mutants die at late embryonic stages due to their inability to inflate their trachea properly (Zhang and Ward, 2009), we utilized CRISPR/Cas9 to generate uif deficient wrapping glial cells to independently verify the RNAi-based phenotype. Four different sgRNAs were generated, targeting different regions of the uif locus (Figure 3—figure supplement 1; see Materials and methods for details). Assuming that Cas9-induced double-stranded breaks will generate deletions or indels that will cause the formation of a correspondingly shortened open reading frame, we anticipate that the sgRNA targeting the second exon generates an almost null situation. In contrast, the sgRNA targeting Cas9 to the presumed cleavage site $uif^{sgRNA\ CS}$ or the one targeting a sequence 5' to transmembrane domain might result in the formation of a secreted and not membrane-bound Uif protein. Finally, the sgRNA that targets sequences coding for the short cytoplasmic domain might allow the formation of an almost intact Uif protein.

When Cas9 is expressed ubiquitously together with any of these four sgRNA constructs in trans to an uif deficiency, development is arrested during early larval stages with defects in tracheal inflation, indicating the functionality of the sgRNA constructs ([w; UAS-Cas9/Df(2L)ED438; da-Gal4/UAS-uif$^{sgRNA}$ $^{X}$], data not shown).

When Cas9 is expressed specifically in wrapping glial cells together with any of the four sgRNA constructs, wrapping glial cells appear thin and patchy, similar to what we noted following silencing of uif expression by RNAi ([+/+; UAS-Cas9/+; nrv2-Gal4,UAS-CD8::mCherry/UAS-uif$^{sgRNA\ X}$], Figure 3). These data indicate that all parts of the protein are required in wrapping glial development, as even induction of a C-terminal mutation impairs morphology.

To further analyze the poorly differentiated wrapping glial cells, we initiated an electron microscopic analysis of nerves in third instar larval filet preparations. We took ultrathin sections 150 µm distal from the ventral nerve cord. As a control genotype, we utilized animals that expressed double-stranded RNA targeting GFP-encoding mRNA in all wrapping glial cells (UAS-GFP$^{dsRNA}$). In cross sections, we found severely reduced glial cell processes (Figure 4A and B). Upon knockdown of uif, the wrapping index drastically decreases from 0.17 to 0.03 (Figure 4—figure supplement 1, n=3 larvae, 5–9 nerves per specimen, p=2.88 × 10$^{-7}$).

To test how uif affects wrapping glial development, we performed overexpression studies. Gain of uif function in wrapping glia caused bulge formation around the wrapping glial nucleus (Figure 4C). Similar to what we had noted following expression of activated Heartless, more distal and proximal parts of the wrapping glia cell remained thin and did not fully develop (Figures 1C and 4C). Subsequent ultrastructural analysis revealed a reduced wrapping index outside of the bulge region where only little glial wrapping is observed and most axons are devoid of any glial cell contact (Figure 4D and Figure 3—source data 1). Within the nerve bulge, an excess of wrapping glial membranes can be seen (Figure 4E and F). These processes fail to wrap individual axons, which results in a significantly decreased wrapping index of 0.08 (Figure 4—figure supplement 1, p=2.56 × 10$^{-9}$, n=3 larvae, 5–6 nerves per specimen). Interestingly, the excess glial cell processes form multilayered membrane stacks (Figure 4F). Thus, heartless appears to direct wrapping glial cell growth, while uif is needed for growth and stabilization of a specific, but still elusive, membrane compartment, possibly matching the apical domain of epithelial cells. To directly test this, we stained for the distribution of PIP2 and PIP3 using specific PH-domain sensors, but in contrast to a differential distribution of PIP2 and PIP3 in ensheathing glia (Pogodalla et al., 2021), we found no specific localization of these membrane lipids in wrapping glia.

## Notch is required for wrapping glial cell differentiation

Importantly, although uif is needed for wrapping glial cell growth, it is not sufficient to instruct the wrapping of axons. Thus, Uif might define the wrapping glial cell interface required to wrap axons, and Uif interacting proteins organize subsequent wrapping. One of these interacting proteins is Notch. Uif can bind Notch and antagonizes the canonical Notch signaling pathway (Loubéry et al., 2014; Xie et al., 2012). Likewise, it has been reported that Notch negatively regulates uif transcription (Djiane

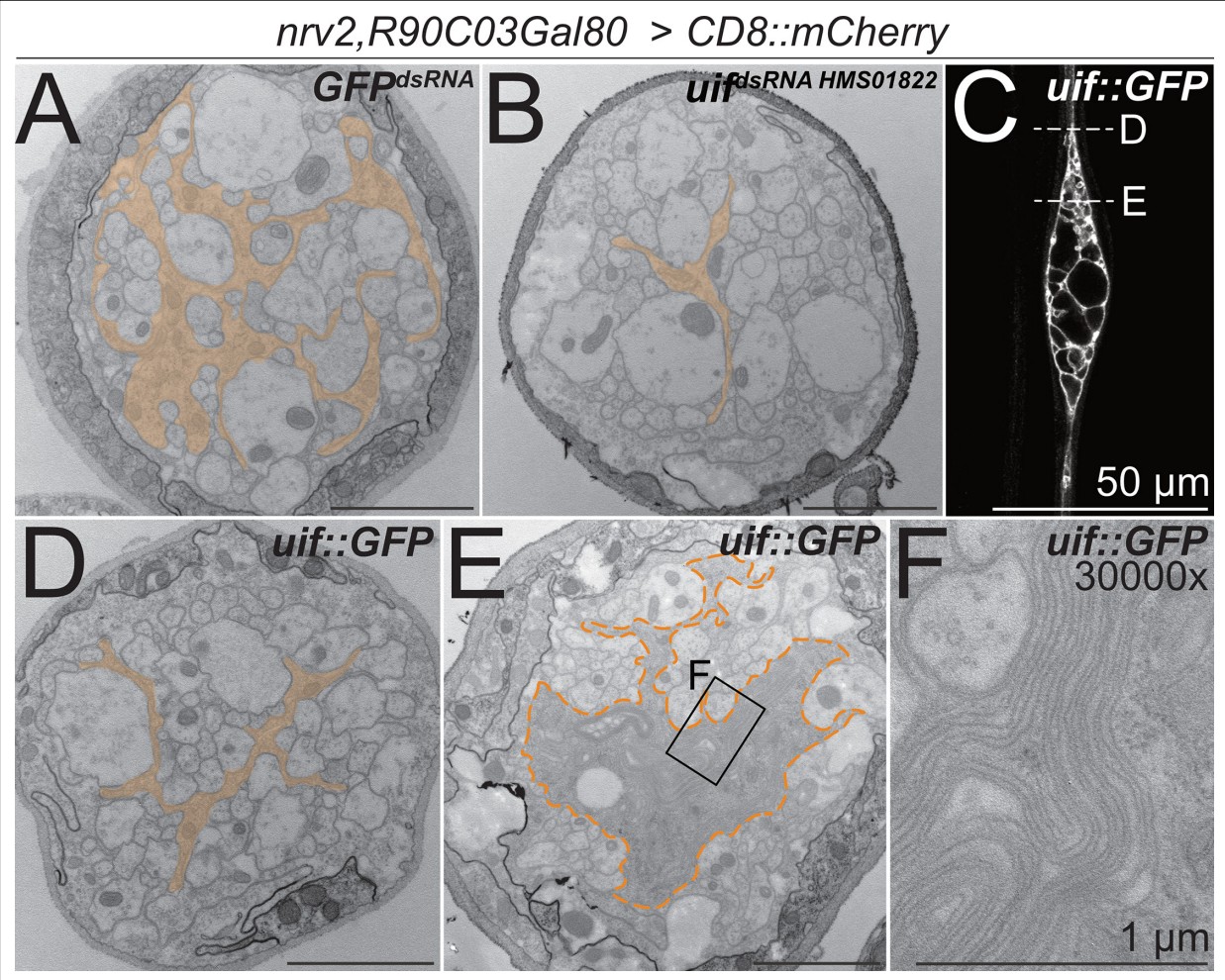

**Figure 4.** *uif* affects axonal ensheathment of wrapping glia. (A+B, D–F) Electron microscopic cross sections of third instar larval abdominal peripheral nerves with wrapping glia-specific expression [*nrv2-Gal4; R90C03-Gal80*] of (**A**) *GFP$^{dsRNA}$* mock control transgene and (**B**) *uif$^{dsRNA-HMS01822}$*. Upon knockdown of *uif*, wrapping glial cell complexity is reduced. (**C**) Single plane of a confocal image of a third instar larval nerve stained for CD8::mCherry expression. Expression of *uif::GFP* in wrapping glial cells causes bulge formation, while outside the bulge, wrapping glia appear thin (arrowheads). The dashed lines indicate the plane of section in relation to the bulge of the images shown in (**D, E**). (**D**) Upon *uif::GFP* overexpression specifically in wrapping glia, glial morphology is reduced outside the bulge region. (**E**) Within the nerve bulge, wrapping glial membrane increases in size while most axons lack proper wrapping. (**F**) Close-up of the region indicated in (**E**). *GFP$^{dsRNA}$* n=4 larvae, 4–7 nerves per specimen; *uif$^{dsRNA}$* n=3 larvae, 5–9 nerves per specimen; *uif::GFP* n=3 larvae, 5–6 nerves per specimen. Scale bars 2 µm unless indicated otherwise.

The online version of this article includes the following figure supplement(s) for figure 4:

**Figure supplement 1.** Quantification of the wrapping index (WI).

*et al., 2013*). Thus, the interaction of Uif and Notch might set a switch triggering glia-axon interaction leading to axon wrapping.

To determine a possible role of *Notch* signaling during wrapping glia differentiation, we first silenced *Notch* expression by RNAi in wrapping glia of otherwise normal animals. Knockdown of *Notch* expression using three different dsRNA constructs targeting different sequences of the *Notch* mRNA specifically in wrapping glia resulted in the appearance of thin wrapping glial cells in larval filet preparations (*Figure 5A–C*). A similar phenotype was also induced by removing *Notch* expression using conditional CRISPR/Cas9-based knockout (*Figure 5D and E*) or in mutant *Notch$^{ts1}$* animals (*Shellenbarger and Mohler, 1975*) that were kept at the restrictive temperature of 29°C during larval stages, only (*Figure 5F and G*). In contrast, upon expression of activated *Notch* (*N$^{ICD}$*), no significant changes of wrapping glial morphology could be detected using the confocal microscope (*Figure 5—figure supplement 1*).

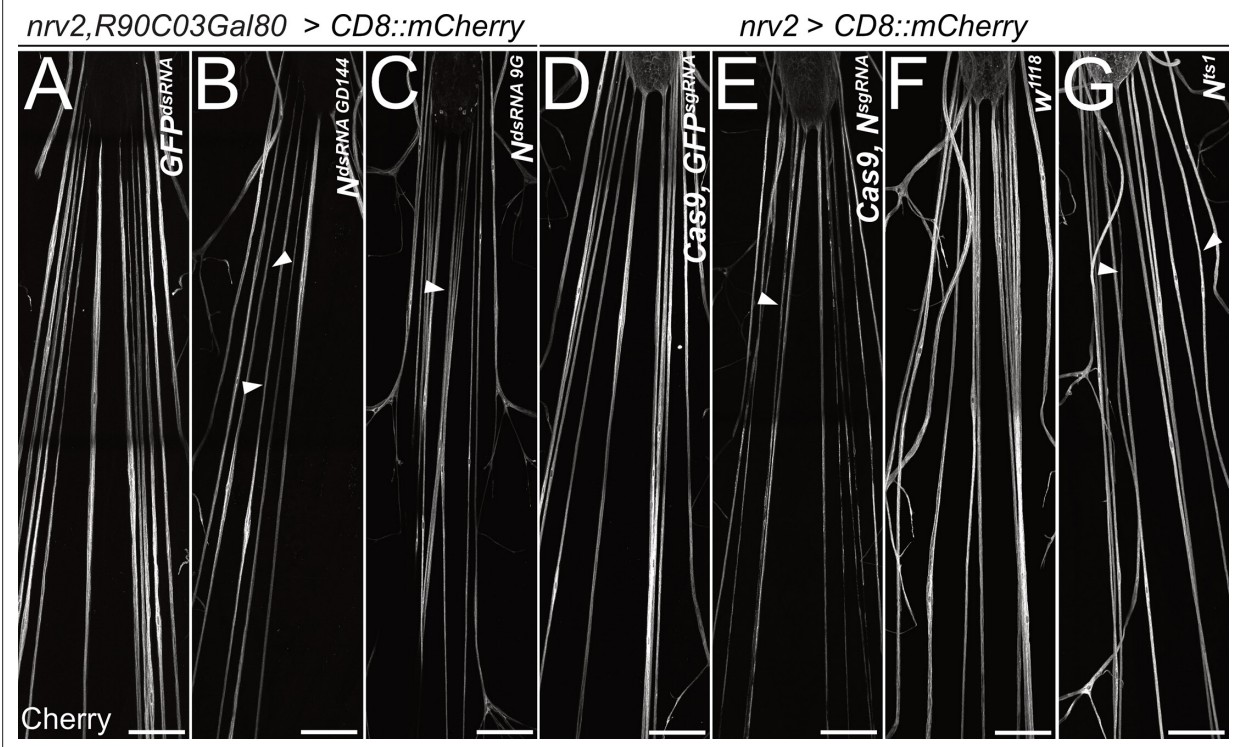

**Figure 5.** *Notch* is required for wrapping glial development. Filet preparations of wandering third instar larvae stained for CD8::mCherry expression. The segmental nerves just posterior to the ventral nerve cord are shown. (**A**) Control larvae expressing the mock control *GFP*<sup>dsRNA</sup> in wrapping glial cells specifically [*nrv2-Gal4; R90C03-Gal80*]. Upon expression of dsRNA targeting *Notch* mRNAs (**B**) $N^{GD144}$, (**C**) $N^{9G}$. (**D**) Control larva expressing sgRNA directed against the GFP open reading frame in [*nrv2-Gal4; UAS-Cas9*]. (**E**) Conditional knockout of *Notch* leads to dramatically altered morphology of wrapping glial cells. (**F**) Control larva cultured at the same temperature regime as the $N^{ts1}$ larva shown in (**G**). Wrapping glial cells appear smaller compared to the control. Scale bars 100 μm.

The online version of this article includes the following figure supplement(s) for figure 5:

**Figure supplement 1.** Notch activation impairs wrapping glial development at higher temperatures.

## *Notch* instructs axonal wrapping

To analyze the role of *Notch* signaling for wrapping glial differentiation in more detail, we performed an electron microscopic analysis. Upon expression of *Notch*<sup>dsRNA</sup>, the wrapping index drops significantly to 0.12 (**Figure 6A, B, and D**, p=0.00079; n=5, larvae with 7–9 nerves per specimen). In contrast, upon expression of the active form of *Notch*, *Notch*<sup>ICD</sup>, we observed a dramatic and significant increase in the wrapping index to 0.28 (**Figure 6C and D**, p=0.014, n=5 larvae with 3–8 nerves per specimen). This indicates that the wrapping glia more efficiently enwraps axons compared to control larvae. Importantly, overexpression of *Notch* does not cause any bulge formation along the nerves as it is noted upon expression of the activated RTK Heartless (**Figure 1C**).

In summary, these data clearly demonstrate that *Notch* is not only required for development of wrapping glial cells, but it is also sufficient to guide axon wrapping as gain of *Notch* function triggers extensive formation of glial processes, resulting in a hyperwrapping phenotype.

## Notch signaling is active in some adult wrapping glial cells

The above data suggest that *Notch* is expressed by differentiated wrapping glia in the PNS. To directly test this, we utilized a CRIMIC-based transposon insertion into the first coding intron of the *Notch* locus (*Notch*<sup>CR00429-TG4.1</sup>) that carries a Trojan Gal4 element, which allows GAL4 expression under control of the endogenous *Notch* promoter (**Diao et al., 2015**; **Lee et al., 2018**). However, no Gal4 activity is associated with the *Notch*<sup>CR00429-TG4.1</sup> insertion. To alternatively detect *Notch* activity, we utilized the common *Notch* activity reporter, *GbeSu(H)-lacZ*, where 3 copies of the Grainy head (Grh) protein binding element (Gbe) and 2 Suppressor of Hairless (Su(H)) protein binding sites drive *lacZ* reporter

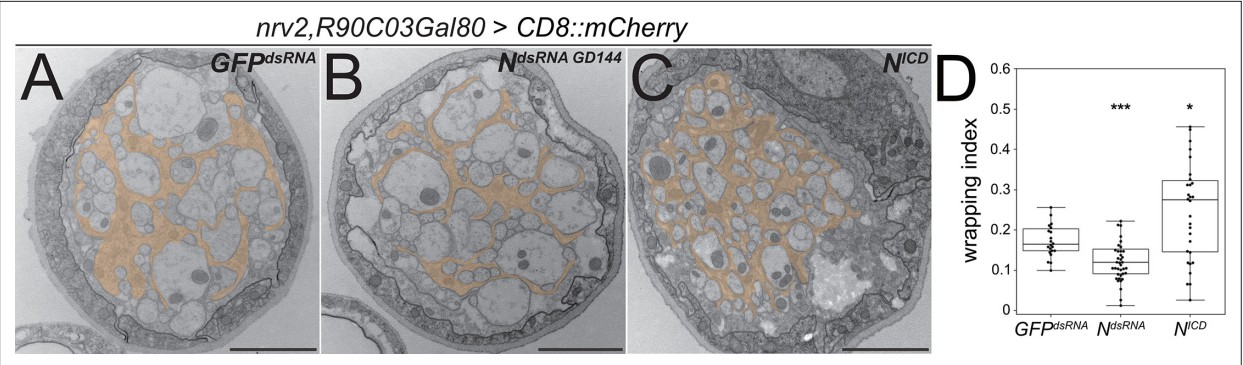

**Figure 6.** *Notch* function affects axonal wrapping. Electron microscopic images of segmental nerves from wandering third instar larvae sectioned 100 μm posterior to the ventral nerve cord. (**A**) Control nerve of an animal expressing *dsRNA* directed against GFP [*nrv2-Gal4; R90C03-Gal80*]. Axons are wrapped by processes of the wrapping glia. The entire nerve is engulfed by the perineurial glia and the subperineurial glia. For quantification of glial wrapping, see (**D**). (**B**) Upon expression of *dsRNA* targeting *Notch mRNA* ($N^{GD144}$), glial wrapping is reduced. (**C**) Upon expression of an activated form of Notch ($N^{intra}$), glial wrapping is increased. (**D**) Quantification of wrapping index (WI). While *Notch* knockdown significantly decreases the WI (from 0.17 to 0.12, p=0.00079), activation of Notch signaling significantly increases glial wrapping and the WI (from 0.17 to 0.27, p=0.014). For statistical analysis, a t-test was performed for normally distributed data (Shapiro test), and Mann-Whitney U test was performed for not normally distributed data. $GFP^{dsRNA}$ n=4 larvae with 4–7 nerves each; $N^{dsRNA}$ n=5, larvae with 7–9 nerves each; $N^{ICD}$ n=5 larvae with 3–8 nerves each. α=0.05, *p≤0.05, ***p≤0.001. Scale bars 2 μm.

The online version of this article includes the following source data and figure supplement(s) for figure 6:

**Source data 1.** Excel file giving the number of axons in the analyzed nerves to calculate the wrapping index.

**Figure supplement 1.** The Notch activity reporter is active in differentiated glia.

expression (**Furriols and Bray, 2001**). In larvae, *Notch* reporter activity was detected in neuroblasts of the brain lobes and the thoracic neuromeres but not in peripheral wrapping glia (**Figure 6—figure supplement 1**). In adults, however, *Notch* activity could be detected in ensheathing glial cells, as well as in peripheral wrapping glial cells (**Figure 6—figure supplement 1**).

The above-mentioned data do not demonstrate whether canonical *Notch* signaling acts in larval wrapping glia. We therefore suppressed expression of the gene *Su(H)*, which encodes a transcription factor critically required for *Notch*-dependent gene expression, specifically in wrapping glia. In such knockdown larvae, overall wrapping glial morphology appeared impaired when analyzed using a confocal microscope (**Figure 7—figure supplement 1**). In an electron microscopic analysis, we could observe a clear reduction in the wrapping index (**Figure 7A and B**, **Figure 4—figure supplement 1**, WI = 0.075, p=4.21 × 10⁻¹¹, n=4 larvae with 6–8 nerves per specimen). This finding indicates an involvement of *Su(H)* in wrapping glia development. Moreover, when we silenced the gene *mastermind* (*mam*), which encodes a further transcription factor involved in *Notch* signaling (**Henrique and Schweisguth, 2019**), we noted a severe impairment of glial morphology at the confocal microscope. Wrapping glial cells accompanying the abdominal nerves appeared thin and not well differentiated (**Figure 7—figure supplement 1**). Similarly, when looking at electron microscopic images, we noted a dramatic reduction in the wrapping index with a corresponding loss of glial complexity (**Figure 7C**, **Figure 4—figure supplement 1**, WI = 0.07, p=4.29 × 10⁻⁹, n=3 larvae with 5–9 nerves per specimen). In conclusion, these data indicate that canonical Notch signaling acts within larval wrapping glial cells to guide engulfment of axons.

### *Notch* activation is not mediated by canonical ligands

Notch is generally activated by the transmembrane EGF-domain proteins encoded by *Delta* or *Serrate*. Both ligands, like the Notch receptor, are evolutionarily well conserved. To determine the expression of both genes, we again utilized insertion of Trojan Gal4 elements in either Delta or Serrate. While Serrate does not appear to be expressed in neurons, we noted some expression of Delta in peripheral sensory neurons and in very few CNS neurons (**Figure 7—figure supplement 2**). We then performed RNAi-based knockdown experiments specifically in neurons using the *nSyb-Gal4* driver and found no effect on wrapping glial morphology at the light microscopic and the electron microscopic level (**Figure 7—figure supplements 2 and 3**, data not shown). These results were confirmed by a second dsRNA construct, as well as by sgRNA/Cas9-mediated, cell type-specific knockout experiments

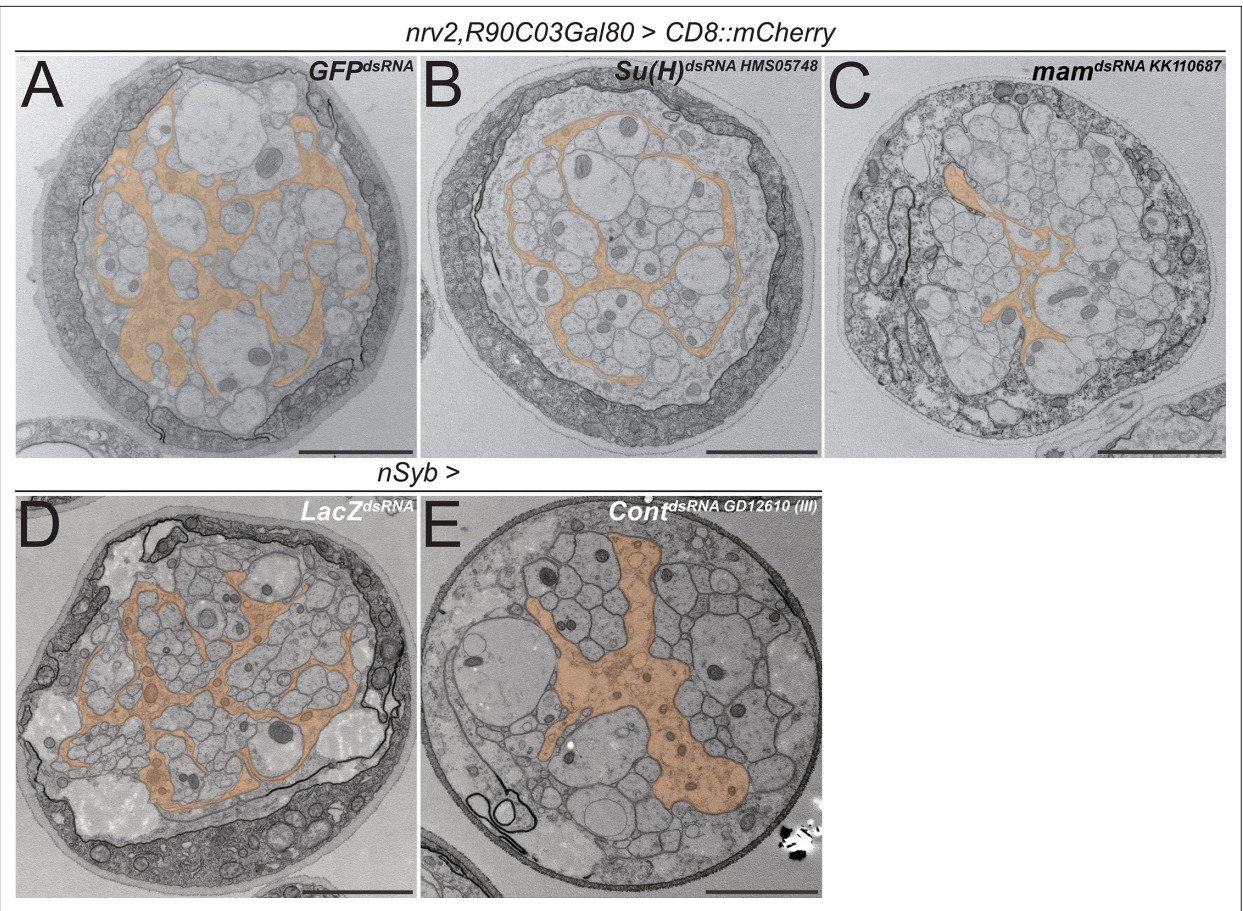

**Figure 7.** Knockdown of *mam*, *Su(H)*, and *contactin* impairs axonal wrapping. Electron microscopic cross sections of third instar larval abdominal peripheral nerves. Glial cell morphology is indicated in orange color. (**A–C**) Wrapping glia-specific [*nrv2-Gal4; R90C03-Gal80*] expression of *GFP*^dsRNA as mock control, (**B**) *Suppressor of Hairless* (*Su(H)*) dsRNA^HMS05748, or (**C**) *mastermind* (*mam*) dsRNA^KK110687. Note reduced complexity of glial cell processes. (**D**) Neuron-specific [*nSyb-Gal4*] expression of *LacZ*^dsRNA as mock control and (**E**) *Contactin* (*Cont*) dsRNA^GD12610 inserted on the third chromosome. Upon neuronal knockdown of *Cont*, glial wrapping of peripheral axons is impaired. Scale bars 2 µm.

The online version of this article includes the following figure supplement(s) for figure 7:

**Figure supplement 1.** Knockdown of *mam* and *Su(H)* impairs wrapping glial development.

**Figure supplement 2.** Neuronal knockdown of Notch ligands *Dl* and *Ser* does not affect wrapping glial morphology.

**Figure supplement 3.** Neuronal knockout of Notch ligands *Dl* and *Ser* does not affect wrapping glial morphology.

(*Figure 7—figure supplement 3*). In summary, these data suggest that during larval development of the peripheral wrapping glia, *Notch* is neither activated by Delta nor by Serrate.

In mice, F3/Contactin, a GPI-linked member of the Ig-domain superfamily, activates Notch during oligodendrocyte differentiation (*Hu et al., 2003*; *Hu et al., 2003*). F3/Contactin is well conserved during evolution and a homolog is encoded in the *Drosophila* genome. *Drosophila* Contactin binds Neuroglian and is an essential component of septate junctions which establish the occluding junctions in all epithelial cells and the glial blood-brain barrier (*Faivre-Sarrailh et al., 2004*; *Izumi and Furuse, 2014*; *Peles and Salzer, 2000*). In addition, single-cell sequencing data indicate that *Contactin* is also expressed by neurons (*Davie et al., 2018*). We therefore silenced *Contactin* expression in all neurons using the neuron-specific driver *nSyb-Gal4*. Such larvae survive, and in the third instar stage, a significant reduction in the wrapping index from 0.15 to 0.11 can be noted (*Figure 7D and E*, *Figure 4—figure supplement 1*), p=0,006, n=5 larvae with 10 nerves per specimen for both RNAi lines. Wrapping glia morphology is impaired, similar to what is noted upon knockdown of *Notch* and its downstream signaling components *Su(H)* and *mam* (*Figures 6 and 7*). This suggests that the non-canonical ligand Contactin acts in both flies and mammals to actively control wrapping glial differentiation and, moreover, indicates that additional ligands may exist to achieve full Notch activation.

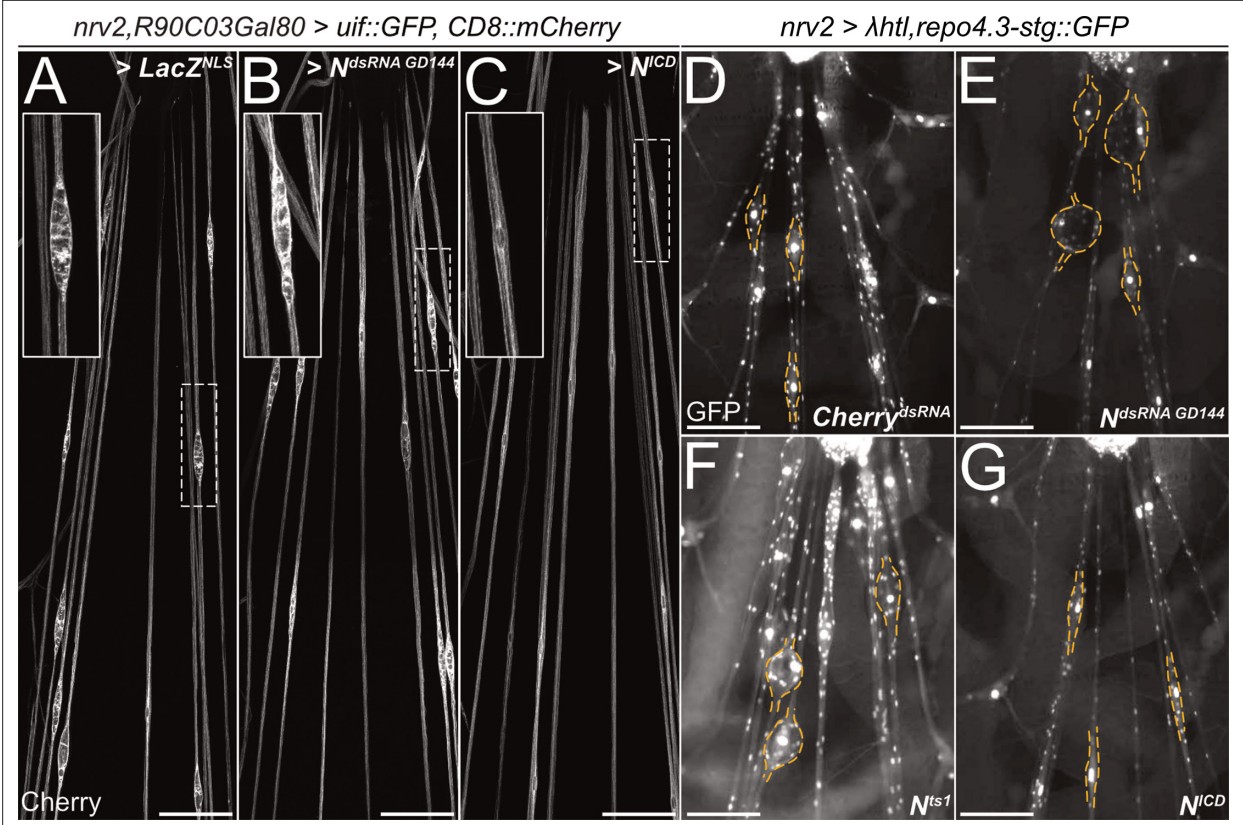

**Figure 8.** *Notch* counteracts *heartless* and *uninflatable* function. Squeeze preparations of wandering third instar larvae. The segmental nerves posterior to the ventral nerve cord are shown. (**A–C**) Larvae expressing *uninflatable* and CD8::mCherry in the wrapping glia. The nerve bulge shown in the white dashed box is enlarged. (**A**) Control larvae co-expressing *LacZ*. Uif expression results in nerve bulges. (**B**) Upon co-expression of *Notch*<sup>dsRNA</sup> (*GD144*), the nerve bulging phenotype is not suppressed. (**C**) Upon activation of Notch signaling by expression of N<sup>ICD</sup>, the nerve bulging phenotype is suppressed. (**D–G**) Larvae expressing an activated Heartless receptor (*λhtl*) in the wrapping glia [*nrv2-Gal4*] carrying a *repo3.4-stinger::GFP* element leading to a panglial nuclear GFP expression. (**D**) Control larvae co-expressing *dsRNA* directed against *mCherry*. Note the prominent nerve bulges. (**E**) Upon co-expression of *Notch*<sup>dsRNA</sup>, the nerve bulging phenotype is enhanced. (**F**) Likewise, the nerve bulging phenotype is enhanced in a *Notch*<sup>ts1</sup> mutant background when the larvae are kept at the restrictive temperature. (**G**) Upon activation of Notch signaling by expression of N<sup>ICD</sup>, the nerve bulging phenotype is significantly rescued. n=7 larvae for all genotypes. (A-C) Scale bars 100 µm, (D-G) Scale bars 75 µm.

The online version of this article includes the following source data for figure 8:

**Source data 1.** Excel file giving the number of axons in the analyzed nerves to calculate the wrapping index.

### *Notch* suppresses *heartless* and *uninflatable* during glial development

To further test whether Notch instructs glial axon wrapping downstream of *heartless* and *uninflatable*, we conducted epistasis experiments. Gain of the *uif* function results in nerve bulges (***Figure 4C***). When we concomitantly silenced *Notch* using RNAi, the nerve bulging phenotype is not changed (***Figure 8A and B***). However, when we co-expressed the intracellular domain of Notch, *UAS-N*<sup>ICD</sup>, resembling the activated form of Notch, the bulging phenotype is rescued (***Figure 8A and C***). These data suggest that similar to the developing wing (***Xie et al., 2012***; ***Xie et al., 2012***), Uninflatable suppresses Notch function.

Expression of a constitutively active FGF-receptor, *λhtl*, in wrapping glial cells results in prominent nerve bulging (***Figure 8D***). Upon co-expression of *Notch* dsRNA, the size of the *λhtl*-induced nerve bulges increases (***Figure 8E***). A similar enhancement of the bulging phenotype was noted when using the temperature-sensitive *Notch*<sup>ts1</sup> allele and an appropriate temperature regime (***Figure 8F***). In contrast, when we co-expressed N<sup>ICD</sup>, we noted a suppression of the severity of the nerve bulging phenotype (***Figure 8G***). Thus, *Notch* function appears to counteract both *uif* and *htl* during glial development.

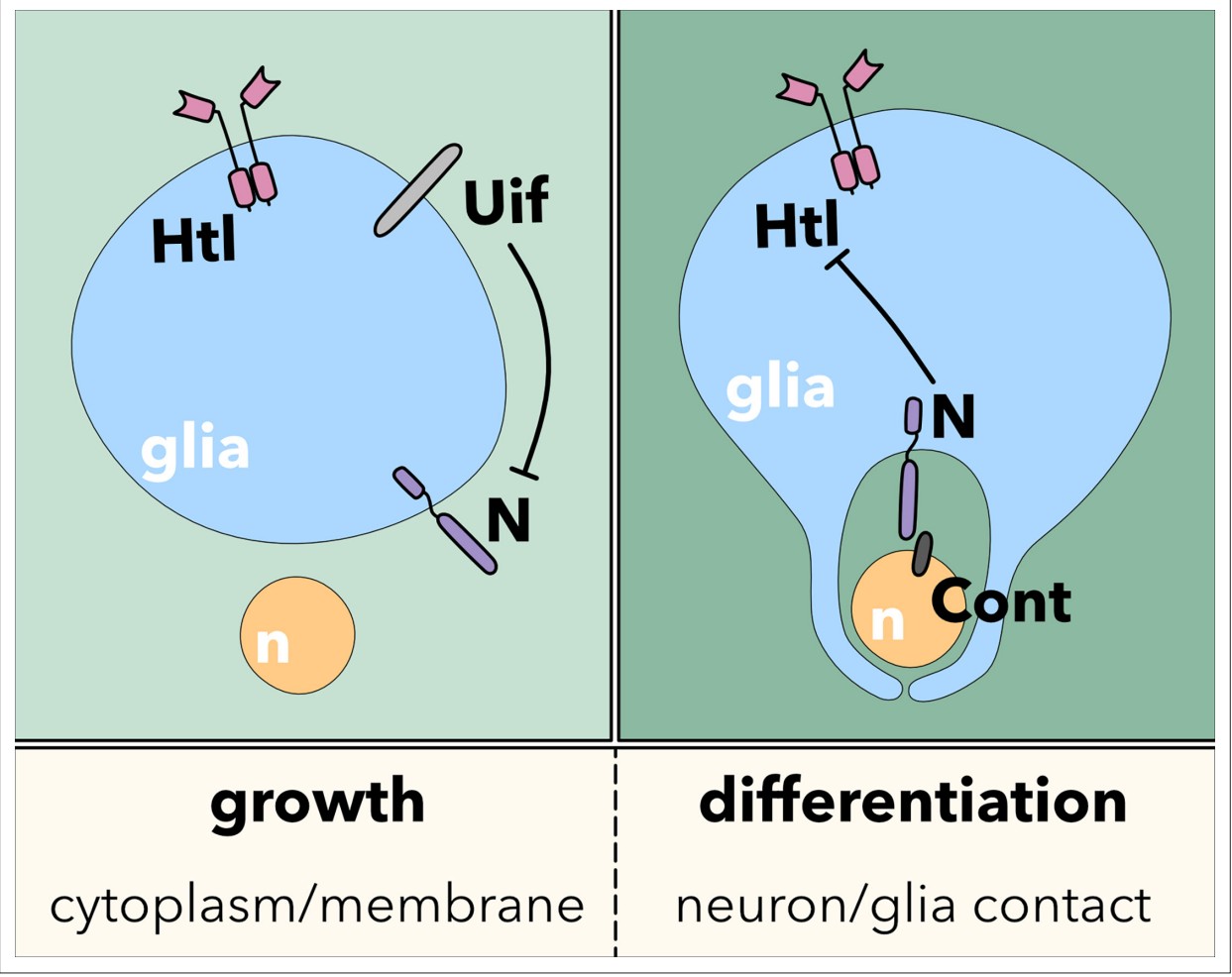

**Figure 9.** Model. The interplay of FGF-receptor, Uninflatable, and Notch signaling controls the switch between glial cell growth and glial cell differentiation, leading to extensive neuron-glia contact. For details, please see the text.

### Model underlying wrapping glia development

We conclude the following model underlying wrapping glia development in the *Drosophila* PNS (*Figure 9*). Initially, the activity of the FGF-receptor Heartless triggers wrapping glial cell growth. The large transmembrane protein Uif is present in the wrapping glia and is needed to stabilize the formation of a specific membrane domain capable of interacting with axons. In addition, Uif suppresses Notch to inhibit precocious axon wrapping. As Heartless negatively regulates expression of Uif (*Avet-Rochex et al., 2014*), Notch becomes more and more active, which also contributes to the silencing of *uif* and *htl*. This then sets the switch to initiate axon wrapping by the wrapping glia (*Figure 9*).

### Discussion

Here, we present a model underlying cell growth and subsequent wrapping of axons by the wrapping glia in the larval PNS of *Drosophila*. In a first step, wrapping glia shows an enormous increase in its size, which is regulated by the RTK Heartless. The large transmembrane protein Uninflatable, which harbors an array of EGF domains in its extracellular domain, is needed to form excessive membrane domains capable of eventually wrapping axons. The last process is controlled by Notch, whose activation then orchestrates axon-glia interaction.

Wrapping glial development is initiated by the activation of RTK, including the FGF-receptor Heartless (Htl). In perineurial glia, RTK activity will trigger cell division, whereas in the remaining glial cells, RTK activity is used to support cell growth to subsequently allow differentiation (*Avet-Rochex et al.,*

*2012*; *Avet-Rochex et al., 2014*; *Franzdóttir et al., 2009*; *Read et al., 2009*; *Stork et al., 2014*; *Witte et al., 2009*; *Wu et al., 2017*). This dual function of RTK signaling in controlling either proliferation, cell growth, or differentiation is accompanied by a switch in activating ligands, as well as the expression of additional regulator proteins that modulate RTK signaling strength (*Franzdóttir et al., 2009*; *Ohm et al., 2024*; *Sieglitz et al., 2013*).

Wrapping glial cells eventually have to cover all axons in the peripheral nerves to prevent degeneration of sensory axons (*Kautzmann et al., 2025*). In *Drosophila* larvae, wrapping glial cells can reach 2 mm in length and thus have an enormous size (*Matzat et al., 2015*). To achieve this, they block mitosis and instead undergo endoreplication (*Unhavaithaya and Orr-Weaver, 2012*; *Von Stetina et al., 2018*; *Zülbahar et al., 2018*). Here, we found that in larval peripheral wrapping glia, Htl supports this cell growth. This is corroborated by our suppressor screen, which identified many genes affecting translation. Similarly, we identified several genes that control metabolite transport across the wrapping glial membrane. Four of these genes are predicted to be involved in sugar/carbohydrate transport (*CG3409, CG4797, CG5078,* and *CG6901*).

Initially, wrapping glial cells form only very thin processes that in peripheral nerves follow the axon bundles (*Matzat et al., 2015*). Starting from second instar larval stages onward, glial cells extend processes that wrap around axon fascicles or single axons (*Kautzmann et al., 2025*; *Matzat et al., 2015*). This process requires the formation of membranes dedicated to axon-glia contact. Possibly, this is mediated by the large transmembrane protein Uninflatable that in epithelial cells is found specifically at the apical plasma membrane and in tracheal cells also can affect polyploidy (*Zhang and Ward, 2009*; *Zhou et al., 2020*).

Using existing antibodies (*Zhang and Ward, 2009*), Uninflatable cannot be detected in third instar larval nerves. Moreover, recent microarray data indicate that activated Heartless suppresses *uif* transcription (*Avet-Rochex et al., 2014*), which suggests that the amount of Uif present in wrapping glia decreases during larval development. Notably, Uif not only induces the generation of numerous membrane sheets, but it can also bind and inhibit Notch function (*Loubéry et al., 2014*; *Xie et al., 2012*). Thus, the inhibition of Notch by Uif is expected to gradually diminish during larval development. This antagonistic action of Uif will stabilize the switch from a glial cell growth phase to a subsequent phase with pronounced axon-glia interaction leading to axon wrapping.

During axon wrapping, Notch likely acts via its canonical signaling cascade. Although mutant analysis and RNAi-based knockdown studies clearly indicate a role of *Notch* in wrapping glia differentiation, we could detect Notch activity only in a subset of adult glial cells. This may be due to the fact that these glial cells form myelin-like structures (*Rey et al., 2023*) and thus require a different level of Notch activity.

Postmitotic functions of Notch have been reported in several instances. In the *Drosophila* nervous system, *Notch* is expressed strongly in axon-associated, postmitotic glial cells (*Allen et al., 2020*; *Davie et al., 2018*; *Li et al., 2022*; *Seugnet et al., 2011*). During embryogenesis, Notch is needed for the exact positioning of glial cells as they migrate along peripheral nerves (*Edenfeld et al., 2007*). Here, we show that in larval wrapping glia, *Notch* instructs axon wrapping. Thus, Notch promotes neuron-glia contact.

How could Notch signaling regulate axon-glia adhesion? The tight interaction of axons and glial cells calls for highly regulated adhesion of the two cell types. The transmembrane proteins of the Ig-domain superfamily Borderless and Turtle bind each other and mediate the differentiation of the wrapping glia in the developing eye (*Cameron et al., 2013*; *Cameron et al., 2016*; *Chen et al., 2017*). While Turtle is expressed on photoreceptor axons, Borderless is found on the wrapping glia. Interestingly, transcriptomic studies have indicated that in the developing eye, Borderless is downstream of Notch (*Nfonsam et al., 2012*).

Given our finding that Notch signaling is acting during axonal engulfment by the wrapping glia, we asked whether one of the canonical Notch ligands, Delta and Serrate (*Henrique and Schweisguth, 2019*), is responsible for Notch activation. However, neither Delta nor Serrate is broadly expressed in neurons, and moreover, neuronal knockdown of neither gene did result in a glial wrapping phenotype. In addition to these canonical ligands, the GPI-linked F3/Contactin protein has been suggested to activate Notch during myelin formation in oligodendrocytes (*Hu et al., 2003*). Here, Notch activation leads to a gamma-secretase-dependent nuclear translocation of Notch[ICD] to upregulate expression of the myelin-related protein MAG and promote myelination. Quite similar, also in

*Drosophila*, knockdown of *contactin* specifically in neurons causes a prominent defect in wrapping glial cell morphology. This surprising evolutionary conservation of the molecular control underlying the differentiation of wrapping glia suggests that *Drosophila* models may be useful to gain a deeper understanding of myelin biology.

# Materials and methods

**Key resources table**

| Reagent type (species) or resource | Designation | Source or reference | Identifiers | Additional information |
|---|---|---|---|---|
| Strain, strain background (*Escherichia coli*) | *E. coli* OneShot TOP10 | Invitrogen | C404003 | Chemically competent cells |
| Genetic reagent (*D. melanogaster*) | UAS-CD8::mCherry | Bloomington Drosophila Stock Center | RRID:BDSC_27391 | |
| Genetic reagent (*D. melanogaster*) | UAS-mCherry$^{dsRNA}$ | Bloomington Drosophila Stock Center | RRID:BDSC_35785 | |
| Genetic reagent (*D. melanogaster*) | w$^{1118}$ | Bloomington Drosophila Stock Center | RRID:BDSC_3605 | |
| Genetic reagent (*D. melanogaster*) | UAS-uif$^{dsRNA\ HMS01822}$ | Bloomington Drosophila Stock Center | RRID:BDSC_38354 | |
| Genetic reagent (*D. melanogaster*) | nSyb-Gal4 | Bloomington Drosophila Stock Center | RRID:BDSC_51635 | |
| Genetic reagent (*D. melanogaster*) | UAS-htl$^{DN}$ | Bloomington Drosophila Stock Center | RRID:BDSC_5366 | |
| Genetic reagent (*D. melanogaster*) | UAS-Cas9 | Bloomington Drosophila Stock Center | RRID:BDSC_58985 | |
| Genetic reagent (*D. melanogaster*) | UAS-Su(H)$^{dsRNA\ HMS05748}$ | Bloomington Drosophila Stock Center | RRID:BDSC_67928 | |
| Genetic reagent (*D. melanogaster*) | nrv2-Gal4 | Bloomington Drosophila Stock Center | RRID:BDSC_6800 | |
| Genetic reagent (*D. melanogaster*) | nrv2::GFP | Bloomington Drosophila Stock Center | RRID:BDSC_6828 | |
| Genetic reagent (*D. melanogaster*) | UAS-N$^{dsRNA\ 9G}$ | Bloomington Drosophila Stock Center | RRID:BDSC_7077 | |
| Genetic reagent (*D. melanogaster*) | Dl$^{MI04868-TG4.1}$ | Bloomington Drosophila Stock Center | RRID:BDSC_77753 | |
| Genetic reagent (*D. melanogaster*) | U6-Dl$^{sgRNA}$ | Bloomington *Drosophila* Stock Center | RRID:BDSC_83095 | |
| Genetic reagent (*D. melanogaster*) | N$^{sgRNA}$ | Bloomington Drosophila Stock Center | RRID:BDSC_84168 | |
| Genetic reagent (*D. melanogaster*) | U6-Ser$^{sgRNS}$ | Bloomington Drosophila Stock Center | RRID:BDSC_84169 | |
| Genetic reagent (*D. melanogaster*) | UAS-GFP$^{dsRNA}$ | Bloomington Drosophila Stock Center | RRID:BDSC_9331 | |
| Genetic reagent (*D. melanogaster*) | UAS-H2B-mRFP | Bloomington Drosophila Stock Center | RRID:BDSC_94270 | |
| Genetic reagent (*D. melanogaster*) | UAS-mam$^{dsRNA\ KK110687}$ | Vienna Drosophila Resource Center | VDRC102091 | |
| Genetic reagent (*D. melanogaster*) | UAS-N$^{dsRNA\ GD144}$ | Vienna Drosophila Resource Center | VDRC1112 | |

*Continued on next page*

*Continued*

| Reagent type (species) or resource | Designation | Source or reference | Identifiers | Additional information |
|---|---|---|---|---|
| Genetic reagent (*D. melanogaster*) | UAS-Ser$^{dsRNA\ GD14442}$ | Vienna Drosophila Resource Center | VDRC27172 | |
| Genetic reagent (*D. melanogaster*) | UAS-Cont$^{dsRNA\ GD12610}$ | Vienna Drosophila Resource Center | VDRC28294 | |
| Genetic reagent (*D. melanogaster*) | UAS-Dl$^{dsRNA\ GD2642}$ | Vienna Drosophila Resource Center | VDRC37287 | |
| Genetic reagent (*D. melanogaster*) | UAS-Cont$^{dsRNA\ GD12610}$ | Vienna Drosophila Resource Center | VDRC40613 | |
| Genetic reagent (*D. melanogaster*) | N$^{[ts]1}$ | **Shellenbarger and Mohler, 1975** | | |
| Genetic reagent (*D. melanogaster*) | UAS-LacZ$^{NLS}$ | **Hummel et al., 2002** | | |
| Genetic reagent (*D. melanogaster*) | UAS-λhtl | **Michelson et al., 1998** | | |
| Genetic reagent (*D. melanogaster*) | UAS-N$^{ICD}$ | Klein, University of Düsseldorf | | |
| Genetic reagent (*D. melanogaster*) | UAS-uif::GFP | Gonzalez-Gaitan, **Loubéry et al., 2014** | | |
| Genetic reagent (*D. melanogaster*) | UAS-LacZ$^{dsRNA}$ | Schirmeier, University of Dresden | | |
| Genetic reagent (*D. melanogaster*) | GFP$^{sgRNA}$ | Schirmeier | | |
| Genetic reagent (*D. melanogaster*) | uif$^{sgRNA\ 2nd\ Exon}$ | This work | | See *Figure 3—figure supplement 1* |
| Genetic reagent (*D. melanogaster*) | uif$^{sgRNA\ CS}$ | This work | | See *Figure 3—figure supplement 1* |
| Genetic reagent (*D. melanogaster*) | uif$^{sgRNA\ TMD}$ | This work | | See *Figure 3—figure supplement 1* |
| Genetic reagent (*D. melanogaster*) | uif$^{sgRNA\ CD}$ | This work | | See *Figure 3—figure supplement 1* |
| Genetic reagent (*D. melanogaster*) | nrv2-Gal4;R90C03-Gal80,UAS-CD8::Cherry | **Kottmeier et al., 2020** | | |
| Genetic reagent (*D. melanogaster*) | Gbe +Su(H)-lacZ | **Furriols and Bray, 2001** | | |
| Genetic reagent (*D. melanogaster*) | repo4.3-stg-GFP | This work | | *Figure 1* |
| Antibody | anti-dsRed 1:1000 | Clontech Labs 3P | 632496 | IF(1:1000) |
| Antibody | anti-β-galactosidase (Mouse monoclonal) | Developmental Studies Hybridoma Bank | 40-1a | IF(1:10) |
| Antibody | anti-Repo (Mouse monoclonal) | Developmental Studies Hybridoma Bank | 8D12 | IF(1:5) |
| Antibody | anti-HRP-DyLight 649 (Goat polyclonal) | Dianova | 123-165-021 | IF(1:500) |
| Antibody | anti-GFP (Rabbit polyclonal) | Invitrogen | A6455 | IF(1:1000) |
| Antibody | anti-mouse 488 (Goat polyclonal) | Invitrogen | A10680 | IF(1:1000) |
| Antibody | anti-mouse 568 (Goat polyclonal) | Invitrogen | A11031 | IF(1:1000) |

*Continued on next page*

*Continued*

| Reagent type (species) or resource | Designation | Source or reference | Identifiers | Additional information |
|---|---|---|---|---|
| Antibody | anti-rabbit 488 (Goat polyclonal) | Invitrogen | A11008 | IF(1:1000) |
| Antibody | anti-rabbit 568 (Goat polyclonal) | Invitrogen | A11011 | F(1:1000) |
| Recombinant DNA reagent | pUAST-dU63gRNA vector carrying a ubiquitous U6:3 promoter | Schirmeier, University of Dresden | | |
| Sequence-based reagent | Uif2ndExon_gRNA2_fw | This work | sgRNA for *uif* cleavage 3' to signal sequence | GTCGTTTCAATATCAAGCACTCGT |
| Sequence-based reagent | Uif2ndExon_gRNA2_rev | This work | sgRNA for *uif* cleavage 3' to signal sequence | AAACACGAGTGCTTGATATTGAAA |
| Sequence-based reagent | UifCS_gRNA2_fw | This work | sgRNA for *uif* cleavage 5' to cleavage site | GTCGTGTTCTGCGTACCTCGGTAG |
| Sequence-based reagent | UifCS_gRNA2_rev | This work | sgRNA for *uif* cleavage 5' to cleavage site | AAATCTACCGAGGTACGCAGAACA |
| Sequence-based reagent | UifTMD_gRNA4_fw | This work | sgRNA for *uif* cleavage 5' to transmembrane domain | GTCGCGCTGTGTGGGCTCCTTTAC |
| Sequence-based reagent | UifTMD_gRNA4_rev | This work | sgRNA for *uif* cleavage 5' to transmembrane domain | AAACGTAAAGGAGCCCACACAGCG |
| Sequence-based reagent | UifCD_gRNA1_fw | This work | sgRNA for cleavage of *uif* cytoplasmic domain | GTCGCTACAATGAAACGTACATGA |
| Sequence-based reagent | UifCD_gRNA1_rev | This work | sgRNA for cleavage of *uif* cytoplasmic domain | AAACTCATGTACGTTTCATTGTAG |
| Software, algorithm | Fiji | *Schindelin et al., 2012* | | |

## STAR methods

### Experimental model and study participant details

All *Drosophila* work was conducted according to standard procedures. All fly stocks were kept at room temperature in plastic vials containing *Drosophila* standard food and dry yeast. Crosses were set up with male and virgin female flies in a ratio of 1:3 and kept at 25°C. Induction of $N^{ts1}$ allele was performed by placing Stage 16 embryos at 29°C.

### Method details

#### sgRNA generation

To generate flies carrying sgRNAs targeted to different regions of the *uif* gene, sgRNA sequences specifically designed for the target gene region of interest were integrated into the pUAST-dU63gRNA vector carrying a ubiquitous U6:3 promoter. To do so, sense and anti-sense oligonucleotides containing the respective sgRNA template sequence ($uif^{2ndExon}$: TTCAATATCAAGCACTCGT; $uif^{CS}$: TGTTCTGCGTACCTCGGTAG; $uif^{TMD}$: CGCTGTGTGGGCTCCTTTAC; $uif^{CD}$: CTACAATGAAACGTACATGA) were phosphorylated, annealed, and ligated into the vector. Flies were tested via single-fly PCR. The position of the different guide RNAs is indicated (***Figure 3—figure supplement 1***).

## Heartless modifier screen

To test whether candidate genes can be linked to fibroblast growth factor (FGF) receptor signaling in wrapping glia, we utilized the nerve bulging phenotype caused by expression of a constitutively active Heartless (*UAS-λhtl*) in wrapping glia (*nrv2-Gal4*). The *repo4.3-stinger::GFP* reporter line labels all glial nuclei independent of *Gal4,* while *nrv2-Gal4* was used to express *UAS-λhtl*. This strain was crossed against a collection of *UAS-dsRNA* lines. For screening, living larvae were mounted as live squeezed preparations to ensure best signal-to-noise ratio . Third instar wandering larvae were collected in ice-cold PBS. With their ventral side up, animals were transferred to a drop of silicon fat (KORALISON, medium viscosity, Kurt Obermeier GmbH) onto a microscope slide. To fix larvae in their position, they were covered with a coverslip and squeezed. Tracheae were oriented toward the microscope slide for visualization of the nervous system. About seven larvae per genotype were assessed using a Nikon fluorescence binocular (AZ-100).

## Immunohistochemistry

For confocal analyses, at least six to ten animals including an equal ratio of both female and male animals were dissected. For experiments using the $N^{ts1}$ allele, only hemizygous males were examined. For larval filet preparations, third instar wandering larvae were collected in ice-cold PBS. The larvae were placed on a silicon pad with their dorsal side facing up and secured at both ends using needles. They were then carefully opened along the dorsal midline using dissection scissors and stretched out with four needles. Gut, fat body, and trachea were removed. For adult brain preparations, adult flies were anesthetized with $CO_2$ and briefly dipped in 70% ethanol. The head capsule was cut open with dissection scissors, and the tissue surrounding the brain was removed using forceps. Legs and wings were excised, and the thorax was opened dorsally. The ventral nerve cord was freed from the surrounding tissue to isolate the sample. After dissection, the samples were fixed by covering them with Bouin's solution for 3 min at room temperature. This was followed by three quick buffer exchanges and three additional washes with PBT lasting 20 min each. Following blocking in 10% goat serum/PBT for 1 hr at room temperature, primary antibodies were applied and incubated overnight at 4°C. Then samples were washed three times with PBT for 20 min each and then incubated with secondary antibodies for 3 hr at room temperature. The tissues were covered with Vectashield mounting solution (Vector Laboratories) and stored at 4°C until imaging using a Zeiss LSM880 Fast-Airyscan microscope. Confocal images were analyzed using Fiji.

## Electron microscopic analysis

For electron microscopy analyses, larvae were dissected in 4% PFA and fixed as filet preparations for 45 min at room temperature, which was followed by fixation in 4% paraformaldehyde (PFA) and 0.5% glutaraldehyde in 0.1 M P-buffer at 4°C overnight. The PFA was replaced by 2% $OsO_4$ in 0.1 M P-buffer for 1 hr on ice (dark). Uranyl acetate staining was performed en bloc using a 2% solution in $H_2O$ for 30 min (dark). Following an EtOH series (50%, 70%, 80%, 90%, and 96%) on ice for 3 min each step, final dehydration was done at room temperature with 2×100% EtOH for 15 min and 2× propylene oxide for 15 min. Following slow epon infiltration, specimens were embedded in flat molds and polymerized at 60°C for 2 days. After trimming, ultrathin sections of segmental nerves about 150 μm distant from the tip of the ventral nerve cord were obtained using a 35° ultra knife (Diatome). Sections were collected on formvar-coated copper grids on which they were left to dry for at least 1 hr prior to imaging, which was performed with a Zeiss TEM 900 at 80 kV in combination with a Morada camera (EMSIS).

## Quantification and statistical analysis

Statistical details of every experiment can be found in the figure legends, with n representing the number of examined animals. Normal distribution of values was performed using the Shapiro-Wilk test. To determine the level of significance, the t-test was applied for normally distributed data, while the Mann-Whitney U test was applied for not normally distributed data. Python was also used to generate all statistics and boxplots. For statistical analyses of EM data, the wrapping index was obtained by putting the number of individual wrapped axons or axon fascicles into relation to the number of all axons. A wrapping index of 1 implies that every single axon of the nerve is individually wrapped. All nerves that contained less than 76 or more than 82 axons were not included in the statistical analysis.

## Acknowledgements

We are thankful to Robert Ward for sending antibodies, Stefanie Schirmeier and Marcos Gonzalez Gaitan for flies and DNA. We are grateful to Elena Rinne for help in generating guide RNA constructs and all our lab colleagues for support and discussions. This work was supported by the Deutsche Forschungsgemeinschaft through funds to CK (SFB 1348 B5).

## Additional information

### Funding

| Funder | Grant reference number | Author |
| --- | --- | --- |
| Deutsche Forschungsgemeinschaft | SFB 1348 B5 | Christian Klämbt |

The funders had no role in study design, data collection and interpretation, or the decision to submit the work for publication.

### Author contributions

Marie Baldenius, Steffen Kautzmann, Formal analysis, Investigation, Visualization, Methodology, Writing – review and editing; Rita Kottmeier, Conceptualization, Data curation, Formal analysis, Investigation, Writing – review and editing; Jaqueline Zipfel, Formal analysis, Investigation, Writing – review and editing; Christian Klämbt, Conceptualization, Formal analysis, Supervision, Funding acquisition, Writing – original draft, Project administration, Writing – review and editing

### Author ORCIDs

Marie Baldenius ⓘ https://orcid.org/0009-0002-3765-9813
Steffen Kautzmann ⓘ https://orcid.org/0009-0003-6678-6608
Christian Klämbt ⓘ https://orcid.org/0000-0002-6349-5800

Reviewer #1 (Public review): https://doi.org/10.7554/eLife.105759.3.sa1
Reviewer #2 (Public review): https://doi.org/10.7554/eLife.105759.3.sa2
Author response https://doi.org/10.7554/eLife.105759.3.sa3

## Additional files

### Supplementary files

Supplementary file 1. Summary of Heartless modifier screen, related to STAR methods>Method details>Heartless modifier screen.

Supplementary file 2. Rescue of the Htl gain of function wrapping glial phenotype by two independent double-stranded RNA (dsRNA) transgenes, related to STAR methods >Method details>Heartless modifier screen (*Figure 3*).

Supplementary file 3. Rescue of Htl gain of function wrapping glial phenotype by two overlapping double-stranded RNA (dsRNA) transgenes, related to STAR methods>Method details>Heartless modifier screen.

MDAR checklist

### Data availability

Figure source data contain the numerical data used to generate Boxplot in *Figure 6D*, *Figure 4—figure supplement 1*, *Figure 7—figure supplement 3*. All Drosophila strains reported are available upon request to CK.

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
