## [Editor Report · eLife Assessment]

This **important** study identifies a new key factor in orchestrating the process of glial wrapping of axons in Drosophila wandering larvae. The evidence supporting the claims of the authors is **convincing** and the EM studies are of outstanding quality. After the revision, the authors have addressed most of the concerns and the manuscript has been significantly improved. Both reviewers have agreed on the significance of the work. The work will be of interest to neuroscientists working on glial cell biology.

---

## [Referee Report · Reviewer #1 (Public review)]

Summary:

A central function of glial cells is the ensheathment of axons. Wrapping of larger-diameter axons involves myelin-forming glial classes (such as oligodendrocytes), whereas smaller axons are covered by non-myelin forming glial processes (such as olfactory ensheathing glia). While we have some insights into the underlying molecular mechanisms orchestrating myelination, our understanding of the signaling pathways at work in non-myelinating glia remains limited. As non-myelinating glial ensheathment of axons is highly conserved in both vertebrates and invertebrates, the nervous system of *Drosophila melanogaster*, and in particular the larval peripheral nerves, have emerged as powerful model to elucidate the regulation of axon ensheathment by a class of glia called wrapping glia. This study seeks to specifically address the question, as to which molecular mechanisms contribute to the regulation of the extent of glial ensheathment focusing on the interaction of wrapping glia with axons.

Strengths and Weaknesses:

For this purpose, the study combines state-of-the-art genetic approaches with high-resolution imaging, including classic electron microscopy. The genetic methods involve RNAi mediated knockdown, acute Crispr-Cas9 knock-outs and genetic epistasis approaches to manipulate gene function with the help of cell-type specific drivers. The successful use of acute Crispr-Cas9 mediated knockout tools (which required the generation of new genetic reagents for this study) will be of general interest to the Drosophila community.

The authors set out to identify new molecular determinants mediating the extent of axon wrapping in the peripheral nerves of third instar wandering Drosophila larvae. They could show that over-expressing a constitutive-active version of the Fibroblast growth factor receptor Heartless (Htl) causes an increase of wrapping glial branching, leading to the formation of swellings in nerves close to the cell body (named bulges). To identify new determinants involved in axon wrapping acting downstream of Htl, the authors next conducted an impressive large-scale genetic interaction screen (which has become rare, but remains a very powerful approach), and identified Uninflatable (Uif) in this way. Uif is a large single-pass transmembrane protein which contains a whole series of extracellular domains, including Epidermal growth factor-like domains. Linking this protein to glial branch formation is novel, as it has so far been mostly studied in the context of tracheal maturation and growth. Intriguingly, a knock-down or knock-out of uif reduces branch complexity and also suppresses htl over-expression defects. Importantly, uif over-expression causes the formation of excessive membrane stacks. Together these observations are in in line with the notion that htl may act upstream of uif.

Further epistasis experiments using this model implicated also the Notch signaling pathway as a crucial regulator of glial wrapping: reduction in Notch signaling reduces wrapping, whereas over-activation of the pathway increases axonal wrapping (but does not cause the formation of bulges). Importantly, defects caused by over-expression of uif can be suppressed by activated Notch signaling. Knock-down experiments in neurons suggest further that neither Delta nor Serrate act as neuronal ligands to activate Notch signaling in wrapping glia, whereas knock-down of Contactin, a GPI anchored Immunoglobulin domain containing protein led to reduced axon wrapping by glia, and thus could act as an activating ligand in this context.

Based on these results the authors put forward a model proposing that Uif normally suppresses Notch signaling, and that activation of Notch by Contactin leads to suppression of Htl, to trigger the ensheathment of axons. While these are intriguing propositions, future experiments will need to conclusively address whether and how Uif could "stabilize" a specific membrane domain capable to interact with specific axons.

Moreover, to obtain evidence for Uif suppression by Notch to inhibit "precocious" axon wrapping and for a "gradual increase" of Notch signaling that silences uif and htl, (1) reporters for N and Htl signaling in larvae, (2) monitoring of different stages at a time point when branch extension begins, and (3) a reagent enabling the visualization of Uif expression could be important next tools/approaches. Considering the qualitatively different phenotypes of reduced branching, compared to excessive membrane stacks close to cell bodies, it would perhaps be worthwhile to explore more deeply how membrane formation in wrapping glia is orchestrated at the subcellular level by Uif.

However, the points raised above remain at present technically difficult to address because of the lack of appropriate genetic reagents. Also more detailed electron microscopy analyses of early developmental stages and comparisons of effects on cell bodies compared to branches will be very labor-intensive, and indeed may represent a new study.

In summary, in light of the importance of correct ensheathment of axons by glia for neuronal function, the proposed model for the interactions between Htl, Uif and N to control the correct extent of neuron and glial contacts will be of general interest to the glial biology community.

Comments on revisions:

The authors have addressed all my comments. However, the sgRNAs in the Star method table are still all for cleavage just before the transmembrane domain, while the Supplemental figure suggests different locations.

---

## [Referee Report · Reviewer #2 (Public review)]

The FGF receptor Heartless has previously been implicated in Drosophila peripheral glial growth and axonal wrapping. Here, the authors performed a large-scale screen of over 2,600 RNAi lines to identify factors regulating the downstream signaling of this process. They identified the transmembrane protein Uninflatable (Uif) as essential for the formation of plasma membrane domains. Furthermore, they found that Notch, a regulatory target of Uif, is required for glial wrapping. Interestingly, additional evidence implies that Notch reciprocally regulates uif and htl, suggesting a feedback loop. Consequently, the authors propose that Uif functions as a 'switch' to regulate the balance between glial growth and axonal wrapping.

Little is known about how glial cell properties are coordinated with axons, and the identification of Uif provides essential insight into this orchestration. The manuscript is well-written, and the experiments are generally well-controlled. The electron microscopy studies, in particular, are of outstanding quality and help mechanistically dissect the consequences of Uif and Notch signaling in the regulation of glial processes. Together, this important study provides convincing evidence of a new player coordinating the glial wrapping of axons.

Comments on revisions:

Overall, the authors have done an excellent job of responding to my substantive concerns in this significantly improved manuscript. In particular, the authors have provided important additional details about the design, prioritization, and outcomes of their screen, and relayed changes that strengthen and extend the impact of their study. I have revised my assessment accordingly, and I expect this study to be of high interest to a variety of researchers in the field.

---

## [Author Response]

The following is the authors’ response to the current reviews.

We would like to proceed with this paper as a Version of Record but we will correct the mistake that we made in the Key resources table. As the reviewer noted we had added the wrong guide RNA sequence here. We are super thankful to the reviewer and apologize for the mistake.

The following is the authors’ response to the original reviews.

**eLife Assessment**
This important study identifies a new key factor in orchestrating the process of glial wrapping of axons in Drosophila wandering larvae. The evidence supporting the claims of the authors is convincing and the EM studies are of outstanding quality.

We are thankful for this kind and very positive judgment.

However, the quantification of the wrapping index, the role of Htl/Uif/Notch signaling in differentiation vs growth/wrapping, and the mechanism of how Uif "stabilizes" a specific membrane domain capable of interacting with specific axons might require further clarification or discussion.

This is now addressed

**Reviewer #1 (Public review):**
Summary:A central function of glial cells is the ensheathment of axons. Wrapping of larger-diameter axons involves myelin-forming glial classes (such as oligodendrocytes), whereas smaller axons are covered by non-myelin-forming glial processes (such as olfactory ensheathing glia). While we have some insights into the underlying molecular mechanisms orchestrating myelination, our understanding of the signaling pathways at work in non-myelinating glia remains limited. As non-myelinating glial ensheathment of axons is highly conserved in both vertebrates and invertebrates, the nervous system of *Drosophila melanogaster*, and in particular the larval peripheral nerves, have emerged as a powerful model to elucidate the regulation of axon ensheathment by a class of glia called wrapping glia. Using this model, this study seeks to specifically address the question, as to which molecular mechanisms contribute to the regulation of the extent of glial ensheathment focusing on the interaction of wrapping glia with axons.Strengths and Weaknesses:For this purpose, the study combines state-of-the-art genetic approaches with high-resolution imaging, including classic electron microscopy. The genetic methods involve RNAi-mediated knockdown, acute Crispr-Cas9 knock-outs, and genetic epistasis approaches to manipulate gene function with the help of cell-type specific drivers. The successful use of acute Crispr-Cas9 mediated knockout tools (which required the generation of new genetic reagents for this study) will be of general interest to the Drosophila community.The authors set out to identify new molecular determinants mediating the extent of axon wrapping in the peripheral nerves of third-instar wandering Drosophila larvae. They could show that over-expressing a constitutive-active version of the Fibroblast growth factor receptor Heartless (Htl) causes an increase in wrapping glial branching, leading to the formation of swellings in nerves close to the cell body (named bulges). To identify new determinants involved in axon wrapping acting downstream of Htl, the authors next conducted an impressive large-scale genetic interaction screen (which has become rare, but remains a very powerful approach), and identified Uninflatable (Uif) in this way. Uif is a large single-pass transmembrane protein that contains a whole series of extracellular domains, including Epidermal growth factor-like domains. Linking this protein to glial branch formation is novel, as it has so far been mostly studied in the context of tracheal maturation and growth. Intriguingly, a knock-down or knock-out of uif reduces branch complexity and also suppresses htl over-expression defects. Importantly, uif over-expression causes the formation of excessive membrane stacks. Together these observations are in in line with the notion that htl may act upstream of uif.Further epistasis experiments using this model implicated also the Notch signaling pathway as a crucial regulator of glial wrapping: reduction in Notch signaling reduces wrapping, whereas over-activation of the pathway increases axonal wrapping (but does not cause the formation of bulges). Importantly, defects caused by the over-expression of uif can be suppressed by activated Notch signaling. Knock-down experiments in neurons suggest further that neither Delta nor Serrate act as neuronal ligands to activate Notch signaling in wrapping glia, whereas knock-down of Contactin, a GPI anchored Immunoglobulin domain-containing protein led to reduced axon wrapping by glia, and thus could act as an activating ligand in this context.Based on these results the authors put forward a model proposing that Uif normally suppresses Notch signaling, and that activation of Notch by Contactin leads to suppression of Htl, to trigger the ensheathment of axons. While these are intriguing propositions, future experiments would need to conclusively address whether and how Uif could "stabilize" a specific membrane domain capable of interacting with specific axons.

We absolutely agree with the reviewer that it would be fantastic to understand whether and how Uif could stabilize specific membrane domains that are capable of interacting with axons. To address this we need to be able to label such membrane domains and unfortunately we still cannot do so. We analyzed the distribution of PIP2/PIP3 but failed to detect any differences. Thus we still lack wrapping glial membrane markers that are able to label specific compartments.

Moreover, to obtain evidence for Uif suppression by Notch to inhibit "precocious" axon wrapping and for a "gradual increase" of Notch signaling that silences uif and htl, (1) reporters for N and Htl signaling in larvae, (2) monitoring of different stages at a time point when branch extension begins, and (3) a reagent enabling to visualize Uif expression could be important next tools/approaches. Considering the qualitatively different phenotypes of reduced branching, compared to excessive membrane stacks close to cell bodies, it would perhaps be worthwhile to explore more deeply how membrane formation in wrapping glia is orchestrated at the subcellular level by Uif.

In the revised version of the manuscript we have now included the use of Notch and RTK-signaling reporters.

(1) reporters for N and Htl signaling in larvae,

We had already employed the classic reporter generated by the Bray lab: Gbe-Su(H)-lacZ. This unfortunately failed to detect any activity in larval wrapping glia nuclei but was able to detect Notch activity in the adult wrapping glia (Figure S5C,F).

We did, as requested, the analysis of a RTK signaling reporter. The activity of sty-lacZ that we had previously characterized in the lab (Sieglitz et al., 2013) increases by 22% when Notch is silenced. Given the normal distribution of the data points, this shows a trend which, however, is not in the significance range. We have not included this in the paper, but would be happy to do so, if requested.

(2) monitoring of different stages at a time point when branch extension begins,

The reviewer asks for an important question; however, this is extremely difficult to tackle experimentally. It would require a detailed electron microscopic analysis of early larval stages which cannot be done in a reasonable amount of time. We have however added additional information on wrapping glia growth summarizing recently published work from the lab (Kautzmann et al., 2025).

(3) a reagent enabling to visualize Uif expression could be important next tools/approaches.

The final comment of the reviewer also addresses an extremely relevant and important issue. We employed antibodies generated by the lab of R. Ward, but they did not allow detection of the protein in larval nerves. We also attempted to generate anti-Uif peptide antibodies but these antibodies unfortunately do not work in tissue. We are still trying to generate suitable reagents but for the current revision cannot offer any solution.

Lastly, we agree with the reviewer that it would be worthwhile to explore how Uif controls membrane formation at the subcellular level. This, however, is a completely new project and will require the identification of the binding partners of Uif in wrapping glia to start working on a link between Uif and membrane extension. The reduced branching phenotype might well be a direct consequence of excessive membrane formation as it likely blocks recourses needed for efficient growth of glial processes.

Finally, in light of the importance of correct ensheathment of axons by glia for neuronal function, this study will be of general interest to the glial biology community.

We are very grateful for this very positive comment.

**Reviewer #2 (Public review):**
The FGF receptor Heartless has previously been implicated in Drosophila peripheral glial growth and axonal wrapping. Here, the authors perform a large-scale screen of over 2600 RNAi lines to find factors that control the downstream signaling in this process. They identify a transmembrane protein Uninflatable to be necessary for the formation of plasma membrane domains. They further find that a Uif regulatory target, Notch, is necessary for glial wrapping. Interestingly, additional evidence suggests Notch itself regulates uif and htl, suggesting a feedback system. Together, they propose that Uif functions as a "switch" to regulate the balance between glial growl and wrapping of axons.Little is known about how glial cell properties are coordinated with axons, and the identification of Uif is a promising link to shed light on this orchestration. The manuscript is well-written, and the experiments are generally well-controlled. The EM studies in particular are of outstanding quality and really help to mechanistically dissect the consequences of Uif and Notch signaling in the regulation of glial processes. Together, this valuable study provides convincing evidence of a new player coordinating the interactions controlling the glial wrapping of axons.
**Reviewer #1 (Recommendations for the authors):**
(1) To be reproducible and understandable, it would be important to provide detailed information about crosses and genotypes, as reagents are currently listed individually and genotypes are provided in rather simplified versions.

We have added the requested information to the text.

(2) Neurons are inherently resistant to RNAi-mediated knockdown and it thus may be necessary to introduce the over-expression of UAS-dcr2 when assessing neuronal requirements and to specifically exclude Delta or Serrate as ligands.

We agree with the reviewer and have repeated the knockdown experiments using UAS-dcr2 and obtained the same results. To use an RNAi independent approach we also employed sgRNA expression in the presence of Cas9. The neuron specific gene knockout also showed no glial wrapping phenotype. These results are now added to the manuscript.

(3) Throughout the manuscript, the authors use the terms "growth" and "differentiation" referring to the extent of branch formation versus axon wrapping. However glial differentiation and growth could have different meanings (for instance, growth could implicate changes in cell size or numbers, while differentiation could refer to a change from an immature precursor-like state to a mature cell identity). It may thus be useful to replace these general terms with more specific ones.

This is a very good point. When we use the term “growth” we only infer on glial cell growth and thus, the increase in cell mass. Proliferation is excluded and this is now explicitly stated in the manuscript. The term “differentiation” is indeed difficult and therefore we changed it either directly addressing the morphology or to axon wrapping.

(4) Page 4. "remake" fibers should be Remak fibers.

We have corrected this typo.

(5) Page 5. "Heartless controls glial growth but does promote axonal wrapping", this sentence is not clear in its message because of the "but".

We have corrected this sentence.

(6) Generally, many gene names are used as abbreviations without introductions (e.g. Sos, Rl, Msk on page 7). These would require an introduction.

All genetic elements are now introduced.

(7) Page 8. When Cas9 is expressed ubiquitously ... It would be helpful to add how this is done (nsyb-Gal4, nrv2-Gal4, or another Gal4 driver are used to express UAS-Cas9, as the listed Gal4 drivers seem to be specific to neurons or glia?).

This now added. We used the following genotype for ubiquitous knockout using the four different uif specific sgRNAs (UAS-uif^sgRNA X^): [w; UAS-Cas9/ Df(2L)ED438; da-Gal4 /UAS-uif^sgRNA X^]. We used the following genotype for a glial knockout in wrapping glia ([+/+; UAS-Cas9/+; nrv2-Gal4,UAS-CD8::mCherry/UAS-uif^sgRNA X^]).

We had previously shown that nrv2-Gal4 is a wrapping glia specific driver in the larval PNS (Kottmeier et al., 2020).

Moreover, the authors mention that "This indicates that a putatively secreted version of Uif is not functional". This conclusion would need to be explained in detail.First, because it requires quite some detective work to understand the panels in Figure 1 on which this statement is based; second, since the acutely induced double-stranded breaks in the DNA and subsequent repair may cause variable defects, it may indeed be not certain what changes have been induced in each cell; and third considering that there is a putative cleavage site, would it be not be expected that the protein is not functional, when it is not cleaved, and there is no secreted extracellular part (unless the cleavage site is not required). The latter could probably only be addressed by rescue experiments with UAS transgenes with identified changes.

We agree with the reviewer. The rescue experiments are unfortunately difficult, since even expression of a full length uif construct does not fully rescue the uif mutant phenotype (Loubéry et al., 2014). We therefore explained the conclusion taken from the different sgRNA knockout experiments better and also removed the statement that secreted Uif forms are non-functional.

In the Star Method reagent table, it is not clear, why all 8 oligonucleotides are for "uif cleavage just before transmembrane domain" despite targeting different locations.

We are very sorry for this mistake and corrected it now. Thank you very much for spotting this.

(8) Page 13. However, we expressed activated Notch,... the word "when" seems to be missing, and it would be helpful to specify how this was done (over-expression of N[ICD]).

We corrected it now accordingly.

(9) To strengthen the point similarity of phenotypes caused by Htl pathway over-activation and Uif over-expression, it would be helpful to also show an EM electron micrograph of the former.

We now added an extensive description of the phenotype caused by activated Heartless. This is shown as new Figure 2.

(10) Figure 4C, the larval nerve seems to be younger, as many extracellular spaces between axons are detected.

This perception is a misunderstanding and we are sorry for not explaining this better. The third instar larvae are all age matched. The particular specimen in Figure 4C shows some fixation artifacts that result in the loss of material. Importantly, however, membranes are not affected. Similar loss of material is also seen in Figure 6C. For further examples please see a study on nerve anatomy by (Kautzmann et al., 2025).

(11) The model could be presented as a figure panel in the manuscript. To connect the recommendation section with the above public review, a step forward could be to adjust the model and the wording in the Result section and to move some of the less explored points and thoughts to the discussion.

We are thankful for this advice and have moved an updated model figure to the end of the main text (now Figure 7).

**Reviewer #2 (Recommendations for the authors):**
(1) Screen and the interest in Uif: Out of the ~62 genes that came out of the RNAi screen, why did the authors prioritize and focus on Uif? What were the other genes that came out of the screen, and did any of those impinge on Notch signaling?

We have now more thoroughly described the results of the screen. We selected Uif as it was the only transmembrane // adhesion protein identified and given the findings that Uif decorate apical membrane domains in epithelial cells, we hoped to identify a protein specific for a similar membrane domain in wrapping glia.

Notch as well as its downstream transcription factors were not included in the initial screen, and were only analyzed, once we had seen the contribution of Notch. Interestingly, here is one single hit in our screen linked to Notch signaling: Gp150. Here however, we have tested additional dsRNA expressing lines and were not able to reproduce the phenotype. This information is added to the discussion.

The authors performed a large-scale screen of 2600 RNAi lines, it seems more details about what came out of the screen and why the focus on Uif would benefit the manuscript.

See above comment.

Relatedly, there would be a discussion of the limitations of the screen, and that it was really a screen looking to modify a gain-of-function phenotype from the activated Htl allele; it seems a screen of this design may lead to artifacts that may not reflect endogenous signaling.

We have now added a short paragraph on suppressor screens, employing gain of function alleles to the introduction.

“In Drosophila, such suppressor screens have been used successfully many times (Macagno et al., 2014; Rebay et al., 2000; Therrien et al., 2000). Possibly, such screens also uncover genes that are not directly linked to the signaling pathway under study but this can be tested in further experiments. Our screen led to the unexpected identification of the large transmembrane protein Uninflatable, which in epithelial cells localizes to the apical plasma membrane. Loss of uninflatable suppresses the phenotype caused by activated RTK signaling. In addition, we find that uif knockdown and uif knockout larvae show impaired glial growth while an excess of Uninflatable leads to the formation of ectopic wrapping membrane processes that, however, fail to interact with axons. uninflatable is also known to inhibit Notch. “

(2) In general this study relies on RNAi knockdown, and is generally well controlled in using multiple RNAi lines giving the same phenotype, and also controlled for by tissue-specific gene knockout. However, there is little in the way of antibody staining to directly confirm the target of interest is lost/reduced, which would obviously strengthen the study.Lacking the tools or ability to assess RNAi efficiency (qPCR, antibody staining), some conclusions need to be tempered. For example, in the experiments in Figure S6 regarding canonical Notch signaling, the authors do not find a phenotype by Delta or Serrate knockdown, but there are no experiments that show Delta or Serrate are lost. Thus, if the authors cannot directly test for RNAi efficiency, these conclusions should be tempered throughout the manuscript.

We agree with the reviewer and now provide information on the use of Dicer in our RNAi experiments and conducted new sgRNA/Cas9 experiments. In addition we tempered our wording stating that Dl and or Ser are still possible ligands.

(3) More description is needed regarding how the authors are measuring and calculating the "wrapping index". In principle, the approach seems sound. However, are there cases where axons are "partially" wrapped of various magnitudes, and how are these cases treated in the analysis? Are there additional controls of previously characterized mutants to illustrate the dynamic range of the wrapping index in various conditions?

This is now explained.

Further, can the authors quantify the phenotypes in the axonal "bulges" in Figures 1, 3, and 5?

This is a difficult question. Although we can easily quantify the number of bulges we cannot quantify the severity of the phenotype as this will require EM analysis. Sectioning nerves at a specific distance of the ventral nerve cord already requires very careful adjustments. Sectioning at the level of a bulge is way more difficult and it is not possible to get the number of sections needed to quantify the bulge phenotype.

The fact is that all wrapping glial cells develop swellings (bulges) at the position of the nucleus. As there are in general three wrapping glial cells per segmental nerve, the number of bulges is three.

(4) It seems difficult to clearly untangle the functions of Htl/Uif/Notch in differentiation itself vs subsequent steps in growth/wrapping. For example, if the differentiation steps are not properly coordinated, couldn't this give rise to some observed differences in growth or wrapping at later stages? I'm not sure of any obvious experiments to pursue here, but at least a brief discussion of these issues in the manuscript would be of use.

We have discussed this in our discussion now more carefully. To discriminate the function of the three genes in either differentiation or in a stepwise mode of growth and differentiation.

When comparing the different loss of function phenotypes they al appear the same, which would argue all three genes act in a common process.

However, when we look at gain of function phenotypes, Htl and Uif behave different compared to Notch. This would favor for two distinct processes.

We have now added activity markers for RTK signaling to directly show that Notch silences RTK activity. Unfortunately we were not able to do a similar reciprocal experiment.

Minor:(1) The Introduction is too long, and would benefit from revisions to make it shorter and more concise.

We have shortened the introduction and hopefully made it more concise.

(2) A schematic illustrating the model the authors propose about Htl, Uif, and Notch in glial differentiation, growth, and wrapping would benefit the clarity of this work.

We had previously added the graphical abstract below that we updated and included as a Figure in the main text.

References

Kautzmann, S., Rey, S., Krebs, A., and Klämbt, C. (2025). Cholinergic and glutamatergic axons differentially require glial support in the Drosophila PNS. Glia. 10.1002/glia.70011.

Kottmeier, R., Bittern, J., Schoofs, A., Scheiwe, F., Matzat, T., Pankratz, M., and Klämbt, C. (2020). Wrapping glia regulates neuronal signaling speed and precision in the peripheral nervous system of Drosophila. Nature communications 11, 4491-4417. 10.1038/s41467-020-18291-1.

Loubéry, S., Seum, C., Moraleda, A., Daeden, A., Fürthauer, M., and González-Gaitán, M. (2014). Uninflatable and Notch control the targeting of Sara endosomes during asymmetric division. Current biology : CB 24, 2142-2148. 10.1016/j.cub.2014.07.054.

Macagno, J.P., Diaz Vera, J., Yu, Y., MacPherson, I., Sandilands, E., Palmer, R., Norman, J.C., Frame, M., and Vidal, M. (2014). FAK acts as a suppressor of RTK-MAP kinase signalling in *Drosophila melanogaster* epithelia and human cancer cells. PLoS Genet 10, e1004262. 10.1371/journal.pgen.1004262.

Rebay, I., Chen, F., Hsiao, F., Kolodziej, P.A., Kuang, B.H., Laverty, T., Suh, C., Voas, M., Williams, A., and Rubin, G.M. (2000). A genetic screen for novel components of the Ras/Mitogen-activated protein kinase signaling pathway that interact with the yan gene of Drosophila identifies split ends, a new RNA recognition motif-containing protein. Genetics 154, 695-712. 10.1093/genetics/154.2.695.

Sieglitz, F., Matzat, T., Yuva-Adyemir, Y., Neuert, H., Altenhein, B., and Klämbt, C. (2013). Antagonistic Feedback Loops Involving Rau and Sprouty in the Drosophila Eye Control Neuronal and Glial Differentiation. Science signaling 6, ra96. 10.1126/scisignal.2004651.

Therrien, M., Morrison, D.K., Wong, A.M., and Rubin, G.M. (2000). A genetic screen for modifiers of a kinase suppressor of Ras-dependent rough eye phenotype in Drosophila. Genetics 156, 1231-1242.